# Mechanistic and evolutionary insights into a type V-M CRISPR–Cas effector enzyme

Satoshi N. Omura[1], Ryoya Nakagawa[1], Christian Südfeld[2],
Ricardo Villegas Warren [2], Wen Y. Wu[2], Hisato Hirano[1], Charlie Laffeber[3],
Tsukasa Kusakizako [1], Yoshiaki Kise[1,4], Joyce H. G. Lebbink [3,5], Yuzuru Itoh[1],
John van der Oost [2]✉ & Osamu Nureki [1]✉

RNA-guided type V CRISPR–Cas12 effectors provide adaptive immunity against mobile genetic elements (MGEs) in bacteria and archaea. Among diverse Cas12 enzymes, the recently identified Cas12m2 (CRISPR–Cas type V-M) is highly compact and has a unique RuvC active site. Although the non-canonical RuvC triad does not permit dsDNA cleavage, Cas12m2 still protects against invading MGEs through transcriptional silencing by strong DNA binding. However, the molecular mechanism of RNA-guided genome inactivation by Cas12m2 remains unknown. Here we report cryo-electron microscopy structures of two states of Cas12m2–CRISPR RNA (crRNA)– target DNA ternary complexes and the Cas12m2–crRNA binary complex, revealing structural dynamics during crRNA–target DNA heteroduplex formation. The structures indicate that the non-target DNA strand is tightly bound to a unique arginine-rich cluster in the recognition (REC) domains and the non-canonical active site in the RuvC domain, ensuring strong DNA-binding affinity of Cas12m2. Furthermore, a structural comparison of Cas12m2 with TnpB, a putative ancestor of Cas12 enzymes, suggests that the interaction of the characteristic coiled-coil REC2 insertion with the protospacer-adjacent motif-distal region of the heteroduplex is crucial for Cas12m2 to engage in adaptive immunity. Collectively, our findings improve mechanistic understanding of diverse type V CRISPR–Cas effectors and provide insights into the evolution of TnpB to Cas12 enzymes.

CRISPR–Cas systems provide adaptive immunity against mobile genetic elements (MGEs) in bacteria and archaea and are divided into two classes (classes 1 and 2) and six types (types I–VI)[1,2]. The class 2 systems include types II, V and VI, in which Cas9, Cas12 and Cas13, respectively, function as effector enzymes responsible for interference against MGEs. Recent studies have identified a dozen functionally divergent type V Cas12 effector proteins[3–11]. Cas12 enzymes associate with a CRISPR RNA (crRNA) or dual RNA guides (crRNA and *trans*-activating crRNA) and cleave double-stranded DNA (dsDNA) targets flanked by a protospacer-adjacent motif (PAM), using a single RuvC nuclease domain. Aside from the RuvC domain, Cas12 effector proteins share little sequence similarity, reflecting their diverse biochemical features. Type V Cas12 effectors reportedly evolved from the IS200/IS605 (IS stands for bacterial insertion sequences) superfamily transposon-associated TnpB protein[2,12,13]. Although the sequences of relatively large Cas12 proteins, such as

[1]Department of Biological Sciences, Graduate School of Science, the University of Tokyo, Tokyo, Japan. [2]Laboratory of Microbiology, Wageningen University and Research, Wageningen, the Netherlands. [3]Department of Molecular Genetics, Oncode Institute, Erasmus MC Cancer Institute, Erasmus University Medical Center, Rotterdam, the Netherlands. [4]Curreio, the University of Tokyo, Tokyo, Japan. [5]Department of Radiotherapy, Erasmus University Medical Center, Rotterdam, the Netherlands. ✉e-mail: john.vanderoost@wur.nl; nureki@bs.s.u-tokyo.ac.jp

**Fig. 1 | Cryo-EM structure of the Cas12m2–crRNA–target DNA ternary complex. a**, Domain structure of Cas12m2. **b**,**c**, Cryo-EM maps (**b**) and structural models (**c**) of the Cas12m2–crRNA–target DNA ternary complex. Zinc and magnesium ions in the TNB and RuvC domains are shown as gray spheres. **d**, Schematic of the crRNA and target DNA. The disordered regions are enclosed by dashed boxes. **e**, Structure of the crRNA-and-target DNA complex.

Cas12a, have weak similarities to the TnpB sequence, five recently identified small Cas12 variants (V-U1 to V-U5) show greater sequence similarities to TnpB and are thought to represent the early stage of evolution from TnpB to the larger Cas12 enzymes[2]. Among the type V-U families, the U2–U4 clusters are classified into the Cas12f subtype, in which the effector proteins function as a dimer, while the

**Table 1 | Data collection, processing, model refinement and validation**

| | Data collection and processing | | |
|---|---|---|---|
| Sample | Cas12m2–crRNA–DNA Full R-loop | Cas12m2–crRNA–DNA Intermediate | Cas12m2–crRNA |
| EMDB ID | EMD-34803 | EMD-34804 | EMD-34824 |
| PDB ID | 8HHL | 8HHM | 8HIO |
| Microscope | | Titan Krios G3i | |
| Detector | | Gatan K3 camera | |
| Magnification | | 105,000 | |
| Voltage (kV) | | 300 | |
| Data-collection software | | EPU | |
| Image-processing package | | cryoSPARC | |
| Electron exposure (e$^-$/Å$^2$) | | 55 | 51 |
| Defocus range (µm) | | −0.8 to −1.6 | |
| Pixel size (Å) | | 0.83 | |
| Symmetry imposed | | $C_1$ | |
| Number of movies | | 3,570 | 8,004 |
| Initial particle images (no.) | | 5,067,096 | 6,034,518 |
| Final particle images (no.) | 182,116 | 180,921 | 115,501 |
| Map resolution (Å) | 2.9 | 3.1 | 3.7 |
| FSC threshold | | 0.143 | |
| Map sharpening $B$ factor (Å$^2$) | −70 | −80 | −170 |
| 3DFSC analysis | | | |
| Global resolution (Å) | 2.94 | 3.14 | 3.91 |
| Sphericity | 0.900 | 0.722 | 0.809 |
| **Model building and refinement** | | | |
| Model composition | | | |
| Protein atoms | 4,630 | 4,036 | 5,660 |
| Nucleic acid atoms | 2,337 | 1,497 | 1,019 |
| Metal ions | 4 | 2 | 2 |
| Model refinement | | | |
| Model-Map CC (CC$_{mask}$/CC$_{box}$/CC$_{peaks}$/CC$_{volume}$) | 0.83/0.77/0.71/0.83 | 0.83/0.81/0.72/0.83 | 0.75/0.74/0.60/0.74 |
| Resolution (Å) by model-to-map FSC, threshold 0.50 (masked/unmasked) | 3.01/3.10 | 3.28/3.36 | 4.15/4.25 |
| Average $B$ factor (Å$^2$) (protein/nucleotide/metal ion) | 90.73/107.33/96.40 | 107.59/129.37/158.47 | 155.66/115.67/156.69 |
| r.m.s. deviations | | | |
| Bond lengths (Å) | 0.004 | 0.004 | 0.005 |
| Bond angles (°) | 0.485 | 0.510 | 0.622 |
| **Validation** | | | |
| MolProbity score | 1.64 | 1.89 | 1.68 |
| CaBLAM outliers (%) | 0.85 | 1.39 | 0.86 |
| Clashscore | 6.49 | 6.39 | 10.33 |
| Rotamer outliers (%) | 2.29 | 3.10 | 0.0 |
| C$_\beta$ outliers (%) | 0 | 0 | 0 |
| EMRinger score | 3.37 | 2.37 | 0.93 |
| Ramachandran plot | | | |
| Favored (%) | 98.64 | 97.06 | 97.17 |
| Allowed (%) | 1.36 | 2.94 | 2.83 |
| Outliers (%) | 0.00 | 0.00 | 0.00 |

EMDB, Electron Microscopy Data Bank; FSC, Fourier shell correlation; PDB, Protein Data Bank; CaBLAM, C-Alpha Based Low-resolution Annotation Method; EMRinger score, an extension of the X-ray crystallography validation tool Ringer in cryo-EM.

U5 cluster is classified as the Cas12k subtype, in which the effector proteins are catalytically inactive.

A recent study demonstrated that the type V-M (formerly type V-U1, clade 2) Cas12m2 from *Mycolicibacterium mucogenicum* (hereafter referred to as Cas12m2 for simplicity) exhibits RNA-guided dsDNA interference activity[11]. Cas12m2 consists of 596 amino acid residues and is much smaller than other Cas12 enzymes, except for Cas12f[7]. Cas12m2 associates with a single crRNA to recognize dsDNA targets containing a 5′-TTN-3′ PAM. Interestingly, Cas12m2 lacks target dsDNA-cleavage activity, probably due to the non-canonical His–Asp–Asp (HDD) catalytic residues in the RuvC domain, rather than the canonical, DNA-cleavable active site Asp–Glu–Asp (DED). Hence, instead of target cleavage, Cas12m2 interferes with invading MGEs through transcriptional silencing by strong DNA binding[11]. However, because of the lack of structural information, the molecular mechanism of the miniature Cas12m2 and the evolutionary relationship between TnpB and Cas12m2 remain unknown.

Here, we report multiple cryo-electron microscopy (cryo-EM) structural states of Cas12m2–crRNA–target DNA ternary complexes and the Cas12m2–crRNA binary complex at overall resolutions of 2.9–3.7 Å. The structures reveal a unique target dsDNA-binding mechanism of Cas12m2, clearly explaining why Cas12m2 is capable of tightly binding target DNA but unable to cleave it. Furthermore, a structural comparison of Cas12m2 with TnpB illuminates their great structural similarities and diversities, providing insights into the evolutionary path from TnpB to Cas12 enzymes.

## Results

### Cryo-EM structure of the Cas12m2–crRNA–target DNA complex

To understand the molecular mechanism of Cas12m2, we analyzed its structure in complex with a 56-nucleotide crRNA (including a 20-nucleotide guide) and its 36-base pair (bp) dsDNA target with a 5′-TTG-3′ PAM by cryo-EM. Using three-dimensional (3D) classification and refinement, we discovered two conformational populations of the Cas12m2 ternary complex at overall resolutions of 2.9 Å (state I) and 3.1 Å (state II) (Fig. 1a–c, Table 1, Extended Data Fig. 1 and Supplementary Table 1). In both states, Cas12m2 binds the crRNA and its dsDNA target at a molar ratio of 1:1:1. Although the full-length crRNA–target DNA heteroduplex is visible in state I, a shorter heteroduplex (approximately 12 bp) is observed in state II. For simplicity, we describe the structural features of state I unless otherwise specified.

The present structures revealed that Cas12m2 adopts a bilobed architecture composed of REC and nuclease (NUC) lobes (Fig. 1a–c and Extended Data Fig. 2a). The REC lobe consists of wedge (WED), REC1 and REC2 domains. The WED domain adopts an oligonucleotide–oligosaccharide-binding fold similar to those of other Cas12 enzymes[14–19] (Extended Data Fig. 2b). The REC1 domain consists of three α-helices, and the REC2 domain has a characteristic coiled-coil structure, forming a distinctive architecture resembling a 'sickle' (Extended Data Fig. 2b). The NUC lobe contains RuvC and target nucleic acid-binding (TNB) domains. The RuvC domain has an RNase H fold, and the His317, Asp485 and Asp579 residues are located at positions similar to those of the well-conserved catalytic Asp, Glu and Asp residues in most other Cas12 enzymes[14–19] (Extended Data Fig. 2c). The TNB

domain is inserted into the RuvC domain and contains an HCCC-type zinc finger, in which a zinc ion is coordinated by His549, Cys552, Cys569 and Cys572 (Extended Data Fig. 2d). The guide RNA–target DNA heteroduplex is accommodated within the positively charged central channel formed by the REC and NUC lobes (Fig. 1c and Extended Data Fig. 2a). The displaced non-target DNA strand (NTS) is visible in our density map and is bound to unique pockets located in the REC and RuvC domains (discussed below).

### crRNA architecture and recognition

The crRNA consists of the 20-nucleotide guide segment (G1 to C20) and the 36-nucleotide repeat-derived scaffold (G(−36) to C(−1)) (Fig. 1d). Nucleotides G1 to U17 in the crRNA and dA1 to dC17 in the target DNA strand (TS) form a 17-bp guide RNA–target DNA heteroduplex (Fig. 1d,e and Extended Data Fig. 2e). Unexpectedly, nucleotides dG(−3) to dC(−1) in the TS do not form base pairs with G18 to C20 in the crRNA and instead rehybridize with dG18* to dC20* in the NTS, suggesting that the 17-bp guide RNA–target DNA heteroduplex represents the optimal length for Cas12m2-mediated DNA recognition (to differentiate between NTS and TS in this study, an asterisk is used to denote the NTS; Fig. 1d,e and Extended Data Fig. 2e).

The crRNA scaffold comprises a pseudoknot (PK) and stem and loop regions. The 5′ end of the crRNA (G(−36) to U(−30)) is disordered (Fig. 1d,e and Extended Data Fig. 2f). Specifically, G(−23) forms a non-canonical base pair with A(−2), and C(−22) to C(−17) form canonical Watson–Crick base pairs with G(−8) to G(−3) to compose the PK structure. The PK coaxially stacks with the stem to form a continuous helix. G(−9) and A(−11) in the loop form hydrogen bonds with the ribose moieties of G(−16) and G(−14), respectively, thereby stabilizing the crRNA scaffold (Extended Data Fig. 3).

The crRNA scaffold is accommodated within the groove formed by the WED and RuvC domains and is mainly recognized by these domains (Fig. 2a,b and Extended Data Fig. 3). Notably, the PK and stem are bound by Arg237 and Arg241 through sugar–phosphate backbone interactions, while the nucleobase of the flipped-out A(−24) is sandwiched by His12 and Arg245 and its ribose moiety hydrogen bonds with Arg245 (Figs. 1d and 2c). The G(−12) in the loop forms a hydrogen bond and stacking interactions with Arg447 and Lys448, thereby stabilizing the crRNA loop region (Fig. 1d and Extended Data Fig. 3). Moreover, the first nucleotide C(−1) and the non-canonical G(−23):A(−2) base pair extensively interact with the WED domain in a base-specific manner. The nucleobase C(−1) forms a base-specific hydrogen bond and stacking interactions with the conserved His269 and Arg270, respectively, while G(−23) stacks with His269, stabilizing the non-canonical G(−23):A(−2) base pair (Fig. 2d). Altogether, the present structure reveals the mechanism of crRNA scaffold recognition by the compact Cas12m2.

### Target DNA architecture and recognition

The guide RNA–target DNA heteroduplex is accommodated within the positively charged central channel formed by the REC1, REC2 and RuvC domains and is recognized by the Cas12m2 protein through interactions with its sugar–phosphate backbone (Fig. 2a,b and Extended Data Fig. 3). Met4 in the WED domain stacks with the first G1:dC17 base pair in the heteroduplex, while Arg301 in the WED domain interacts with the

**Fig. 2 | crRNA and target DNA recognition. a**, Recognition sites of the crRNA scaffold and target DNA. **b**, Electrostatic surface potential of the Cas12m2–crRNA–target DNA ternary complex. The crRNA–target DNA heteroduplex is accommodated within a positively charged central channel formed by WED, REC1, REC2 and RuvC domains, and the PAM duplex is sandwiched by the WED and REC1 domains. **c–g**, Recognition of the stem (**c**), the PK (**d**), the crRNA–target DNA heteroduplex (**e**), the rehybridized DNA duplex (**f**) and the PAM duplex (**g**). Hydrogen bonds are depicted with green dashed lines. **h**, DNA binding of Cas12m2 mutants in vitro, measured by SPR spectroscopy, relative to wild type. Values shown are the mean ± s.e.m. of three independent experiments.

Statistical analysis was carried out using Tukey's post hoc test in case of a significant two-way ANOVA result. Significant differences relative to wild-type Cas12m2 are indicated by asterisks. ***$P < 0.001$. Triple, Cas12m2^R111A/R112A/R126A triple mutant. **i**, Unique Arg-rich pocket formed by the REC1 and REC2 domains. Hydrogen bonds are depicted with green dashed lines. **j**, Close-up view of the non-canonical RuvC active site. The magnesium ion coordinated with the NTS is shown as a gray sphere. The cryo-EM density for the RuvC conserved residues, the magnesium ion and the displaced NTS (dG16* and dT17*) is shown as a blue mesh. **k**, Schematic of the non-canonical RuvC active site.

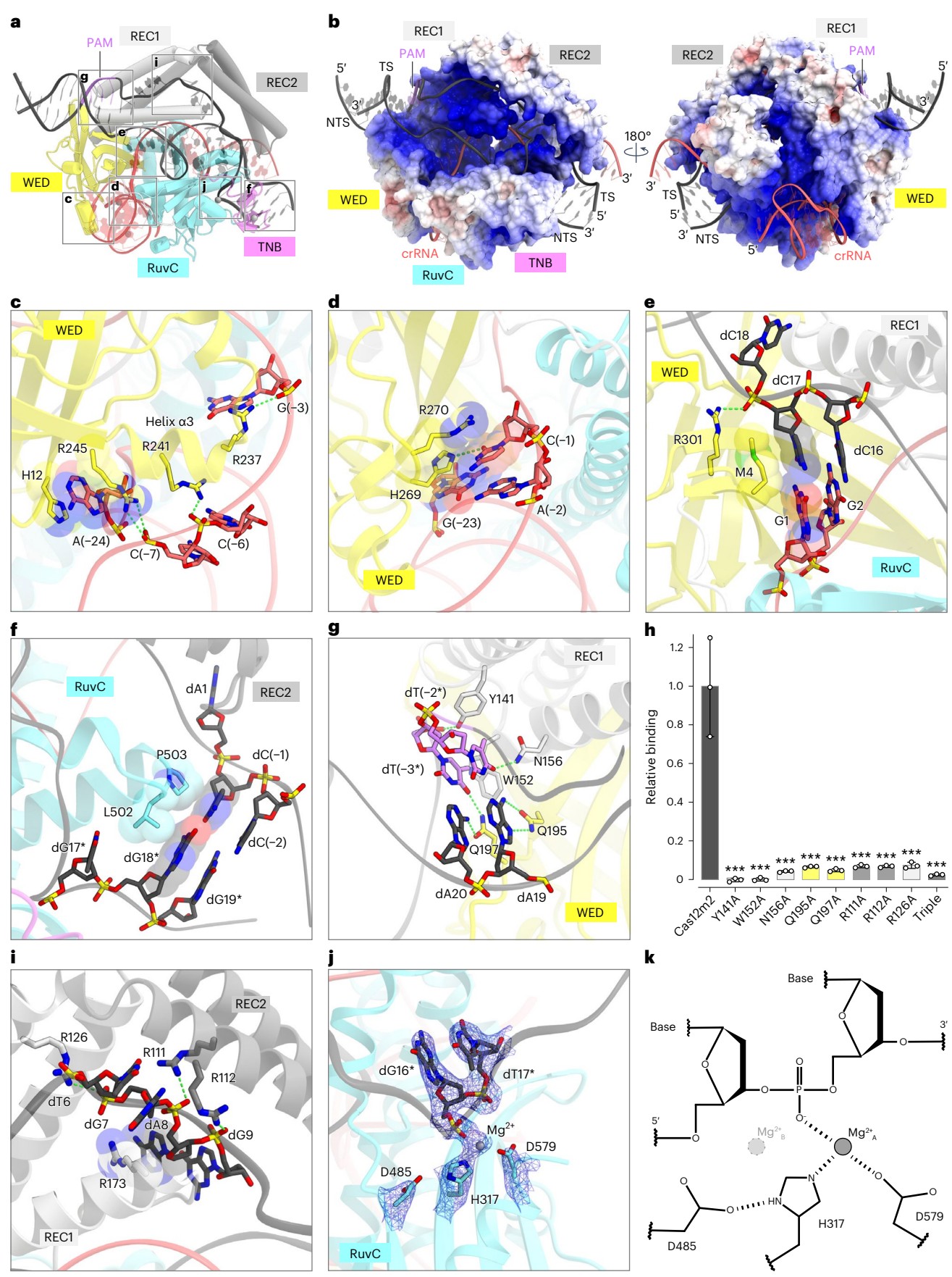

backbone phosphate group of dC18, thereby facilitating heteroduplex formation as observed in other Cas12 enzymes[20] (Fig. 2e). Leu502 and Pro503 in the RuvC domain form hydrophobic interactions with the dG18* and dC(−1) bases, respectively, stabilizing reformation of the DNA duplex downstream of the heteroduplex (Fig. 2f).

The PAM-containing DNA duplex (referred to as the PAM duplex) is distorted into a narrow minor groove and is sandwiched by the REC1 and WED domains (Fig. 2a). The dT(−3*) and dT(−2*) nucleotides form hydrophobic interactions with Trp152 and Tyr141 and base-specific hydrogen bonds with Gln197 and Asn156, respectively (Fig. 2g). Tyr141 also interacts with the phosphate backbone of dT(−3*) (Fig. 2g). The dA19 and dA20 nucleobases, which base pair with dT(−2*) and dT(−3*), form hydrogen bonds with Gln195 and Gln197, respectively (Fig. 2g). Substitution mutagenesis of Cas12m2 confirmed the importance of Tyr141, Trp152, Asn156, Gln195 and Gln197 for DNA binding, as alanine substitution of any of these residues strongly affected the ability of Cas12m2 to interact with its DNA target in vitro (Fig. 2h and Extended Data Fig. 4). These structural and biochemical features explain the requirement of the 5′-TTN-3′ PAM sequence for Cas12m2-mediated target DNA binding.

The displaced NTS passes through the groove formed by the REC1 and REC2 domains toward the groove formed by the RuvC and TNB domains (Fig. 2a,b). Notably, the central region of the negatively charged backbone of the NTS is bound to an Arg-rich pocket in the REC1 and REC2 domains and interacts extensively with Arg111, Arg112, Arg126 and Arg173 (Fig. 2i). Such an Arg-rich pocket has not been observed in any other Cas12 enzymes, even though these Arg residues are highly conserved among Cas12m family enzymes (Extended Data Fig. 5). Cas12m2 mutants with alanine substitutions of Arg111, Arg112 and Arg126 showed reduced binding to target DNA in vitro (Fig. 2h and Extended Data Fig. 4c), suggesting the critical role of the unique Arg-rich pocket for the robust DNA-binding affinity of Cas12m enzymes.

## Non-canonical HDD-type RuvC active site

In most Cas12 enzymes, the RuvC active site has the conserved DED catalytic residues that coordinate two magnesium ions on either side of the scissile phosphate (Extended Data Fig. 6a). The first magnesium ion (A) activates the nucleophilic water molecule, while the second magnesium ion (B) stabilizes the transition state, resulting in a two-divalent cation-dependent DNA-hydrolysis mechanism[21,22] (Extended Data Fig. 6a,b).

In the Cas12m2 structure, the nucleotides dG14* to dT17* in the NTS are clearly visible, and the phosphate group of dG16* is located at the RuvC active site formed by His317, Asp485 and Asp579 residues (Fig. 2j and Extended Data Fig. 6c). At the position corresponding to magnesium ion (A) in other Cas12 enzymes, we observed a magnesium ion that was coordinated by His317 and Asp579, whereas no density corresponding to magnesium ion (B) was observed (Fig. 2j,k and Extended Data Fig. 6c,d). This is due to the replacement of the first Asp and the second Glu in the active site with His317 and Asp485, respectively, in Cas12m2. The His317 side chain has only one lone pair for metal coordination,

which is used to bind magnesium ion (A), and the Asp485 side chain is located too far away to coordinate the second magnesium ion (B) (Fig. 2j,k and Extended Data Fig. 6c,d). Therefore, the non-canonical HDD motif leads to the absence of the second magnesium ion, resulting in loss of cleavage activity for the target DNA. Nonetheless, His317 and Asp579 are bound to the NTS via magnesium ion coordination, and His317 is stabilized by hydrogen bonding with Asp485, suggesting that the HDD motif plays a crucial role in target DNA binding, rather than target DNA cleavage. Consistent with these structural observations, our previous study showed that alanine substitution of Asp485 (HDD > HAD) leads to reduced DNA-binding affinity in vitro and transcriptional silencing activity in vivo[11], indicating that Cas12m2 uses the RuvC catalytic center as the DNA-binding pocket for robust DNA-binding affinity.

## Conformational changes upon heteroduplex formation lead to robust DNA binding

In state I, the target DNA forms a full R-loop containing a 17-bp heteroduplex, whereas, in state II, the target DNA only forms a 12-bp heteroduplex, and the displaced NTS is disordered (Fig. 3a,b). Thus, we propose that state II represents the intermediate state during formation of the 17-bp heteroduplex. Hereafter, we refer to state I as the full R-loop state and to state II as the intermediate state.

Structural comparisons between the full R-loop and the intermediate states revealed conformational changes in the REC2 domain (Fig. 3a–c and Supplementary Video 1). In the intermediate state, the REC2 domain is disordered due to its flexibility (Fig. 3a). By stark contrast, in the full R-loop, the REC2 domain is ordered and interacts with the sugar–phosphate backbone of the heteroduplex at the PAM-distal region (Fig. 3b). This structural rearrangement of the REC2 domain facilitates formation of the Arg-rich pocket between the REC1 and REC2 domains, allowing the binding of the displaced NTS (Fig. 3b). Structural comparisons also indicated local conformational changes in two short α-helices (Lα1 and Lα2) and the loop connecting them (together referred to as the lid motif) in the RuvC domain (Fig. 3c–f). In the intermediate state, residues 499–506 (part of Lα2 and the loop) are disordered due to the flexibility of this region, and Lα1 occludes the RuvC active site (Fig. 3d,f). By contrast, upon formation of the full R-loop, Thr502 in the lid motif hydrogen bonds with the phosphate backbone of dC2, while Pro503 and Leu504 in Lα2 form hydrophobic interactions with dC(−1) and dG18*, respectively (Fig. 3e). These structural rearrangements facilitate the local structural transition of Lα1, opening the RuvC active site to bind the NTS, as observed in Cas12b and Cas12i[23,24] (Fig. 3e,f). Taken together, the conformational changes of the REC2 domain and the lid motif, which enable recognition of the PAM-distal end, provide a binding site for NTS within the Arg-rich pocket and the RuvC DNA-binding site, allowing Cas12m2 to silence transcription of target genes through its robust DNA-binding affinity.

To biochemically validate the importance of the two NTS-binding sites (that is, the Arg-rich pocket and the RuvC active site) in Cas12m2-mediated transcriptional repression, we performed an in vivo transcriptional silencing assay. We generated a target plasmid

**Fig. 3 | Conformational changes upon heteroduplex formation enable robust DNA binding of Cas12m2. a,b**, Structural models of the Cas12m2–crRNA–target DNA ternary complex, representing the intermediate state (**a**) and full R-loop state (**b**). The disordered regions are indicated as dashed lines. **c**, Superimposition of the intermediate state (light blue) and the full R-loop state (colored as in **b**). **d,e**, Close-up views around the lid motif of the intermediate state (**d**) and the full R-loop state (**e**). The disordered regions are indicated as the gray cycle surrounded by dashed lines. Hydrogen bonds are depicted with green dashed lines. **f**, Superimposition of the lid motif of the intermediate state (light blue) and the full R-loop state (colored as in **b**). The displaced NTS (dC15*–dT17*) in the full R-loop state clashes with the Lα2 helix in the intermediate state. **g**, Schematic of the bicistronic *rfp-gfp* operon (pTarget-operon), including the

crRNA targeting sites in the promoter (A1) and at the end of the operon (3′-UTR; E1) described previously[11]. **h**, Normalized RFP and GFP fluorescence of *E. coli* cultures expressing different Cas12m2 mutants, directed to either the promoter (pCRISPR-A1) or the 3′-UTR (pCRISPR-E1). Fluorescence is shown relative to the average fluorescence of cultures expressing a non-targeting guide (pCRISPR-NT). Values shown are the mean ± s.e.m. of two values of biological duplicates (after averaging technical triplicates). Statistical analysis was carried out using Tukey's post hoc test in case of a significant two-way ANOVA result. Significant differences relative to Cas12m2 values are indicated by asterisks. *P < 0.05, **P < 0.01, ***P < 0.001. dCas12m2, Cas12m2^D485A mutant; triple, Cas12m2^R111A/R112A/R126A triple mutant.

(pTarget-operon) containing a bicistronic operon with two fluorescence reporter genes, *rfp* and *gfp*. *Escherichia coli* cells harboring pTarget-operon and either pCRISPR-A1 (crRNA targeting the promoter region of the operon), pCRISPR-E1 (crRNA targeting the 3′ untranslated region (UTR) of the operon) or pCRISPR-NT (crRNA that does not target the operon) were made chemically competent and transformed with either pCas-Cas12m or plasmids encoding its mutants (including

Cas12m2[D485A], Cas12m2[R111A], Cas12m2[R112A] and Cas12m2[R126A] mutants and the Cas12m2[R111A/R112A/R126A] triple mutant) (Fig. 3g). Wild-type Cas12m2 efficiently reduced red fluorescent protein (RFP) and green fluorescent protein (GFP) fluorescence upon crRNA targeting both the A1 and E1 sites, as observed in a previous study (Fig. 3h). By contrast, all mutants exhibited reduced transcriptional silencing capabilities when targeting the E1 site, and the Cas12m2[R111A/R112A/R126A] triple mutant showed

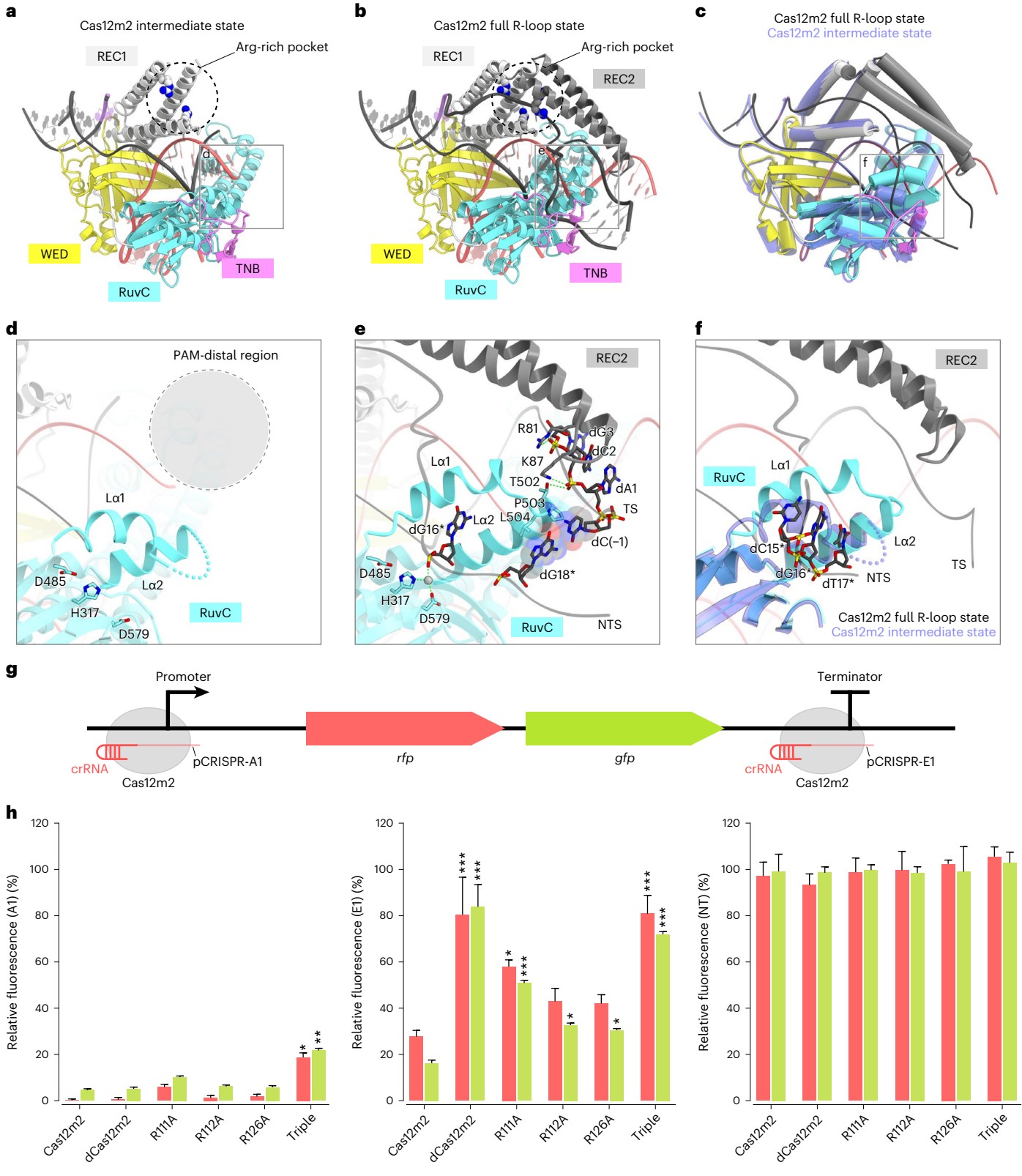

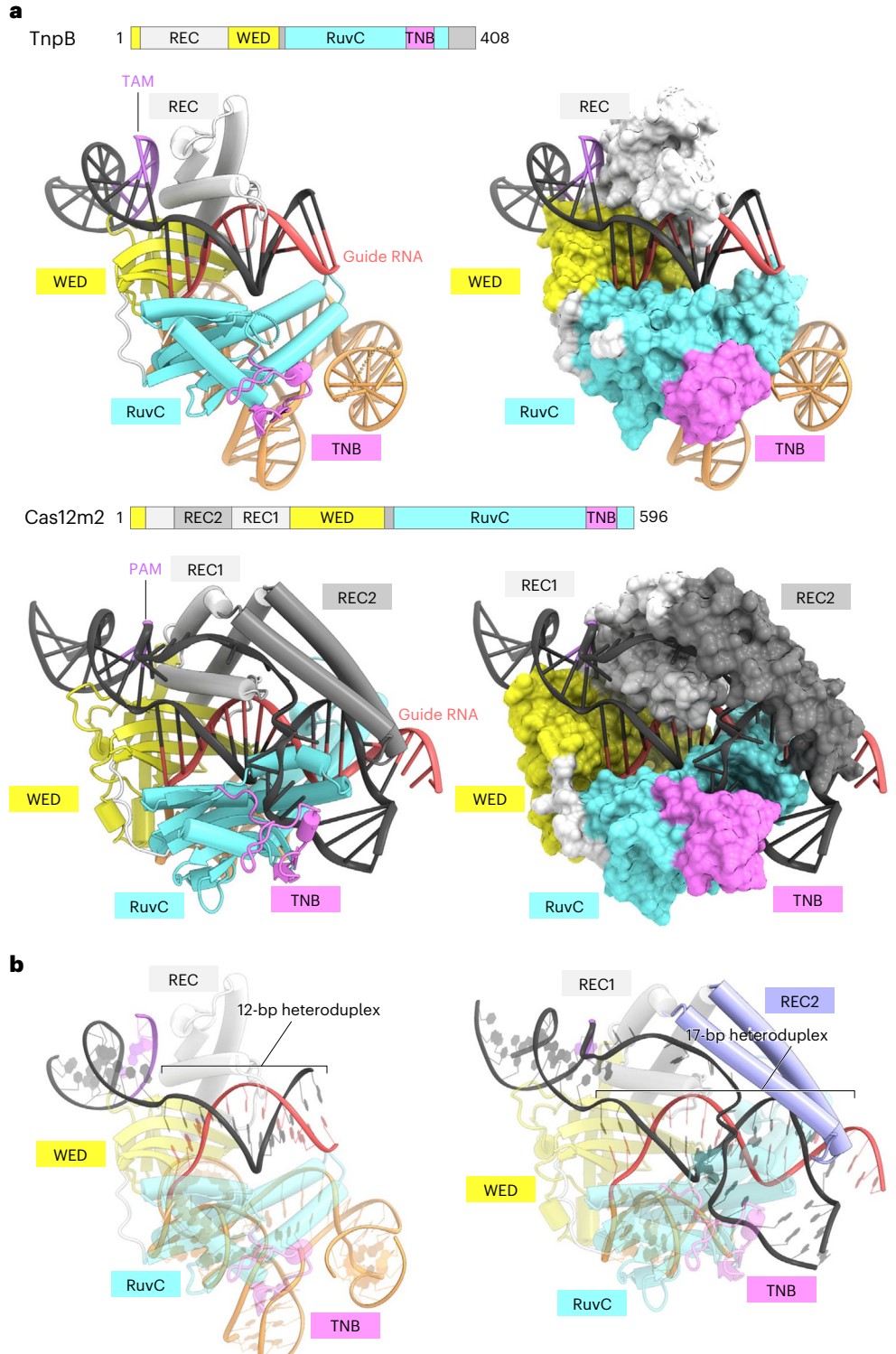

**Fig. 4 | Structural comparison of Cas12m2 and TnpB. a**, Structural comparison of Cas12m2 and TnpB (from *Deinococcus radiodurans* ISDra2) (PDB 8H1J). **b**, Structural comparison of Cas12m2 and TnpB with emphasis on the guide RNA–target DNA heteroduplex. While Cas12m2 shares high sequence similarity with TnpB, Cas12m2 has an inserted REC2 domain (highlighted in blue), which is not present in TnpB, thereby facilitating recognition of the longer guide RNA–target DNA heteroduplex.

lower transcriptional silencing efficacy when targeting both the A1 and E1 sites (Fig. 3h). These results explain why the Arg-rich pocket and the RuvC DNA-binding site, which result from conformational changes of the REC2 domain and the lid motif, play critical roles in Cas12m2-mediated transcriptional silencing.

## Evolutionary path from TnpB to Cas12m2

The Cas12m family enzymes are thought to represent the early stage of evolution from TnpB to larger Cas12 enzymes[2] (Extended Data Fig. 7). Thus, structural comparisons of TnpB and Cas12m2 may provide clues to trace the evolutionary path from TnpB to Cas12 enzymes.

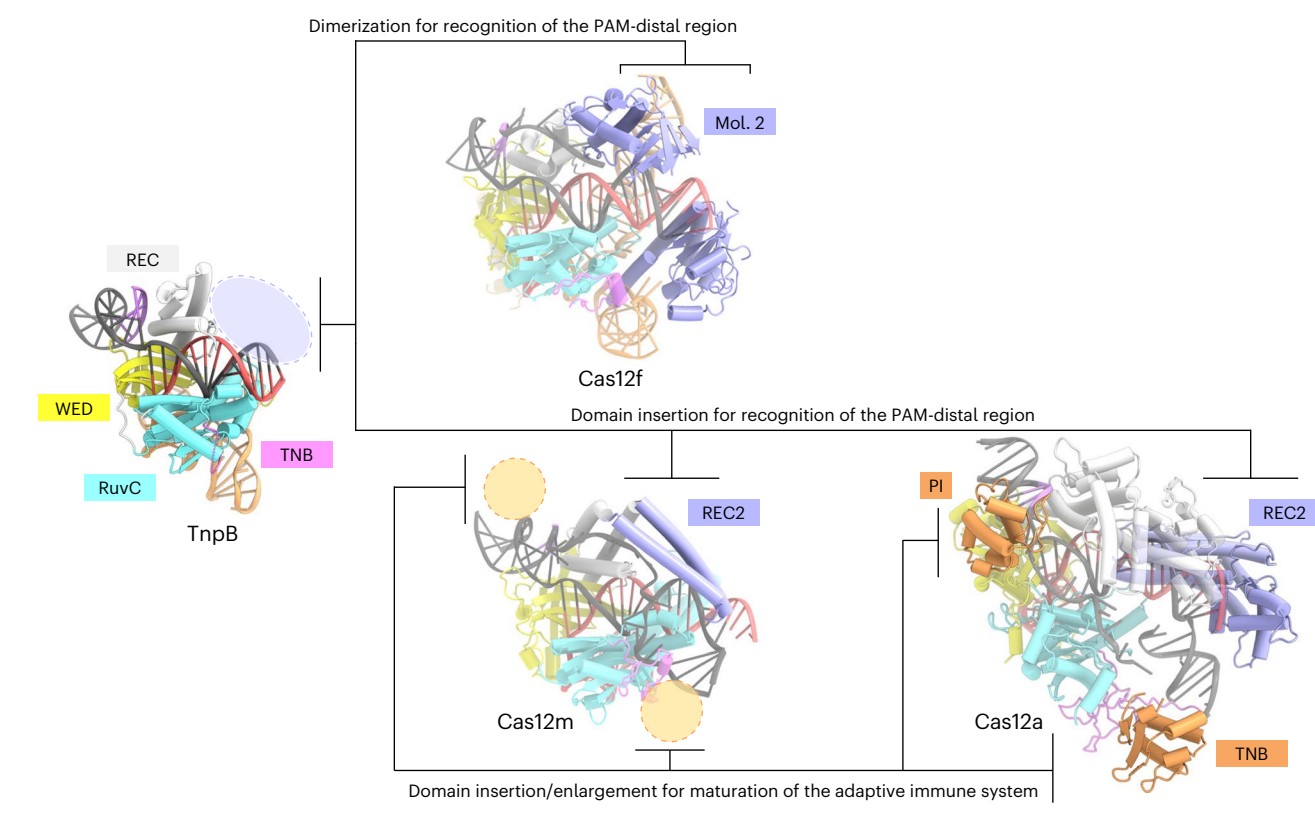

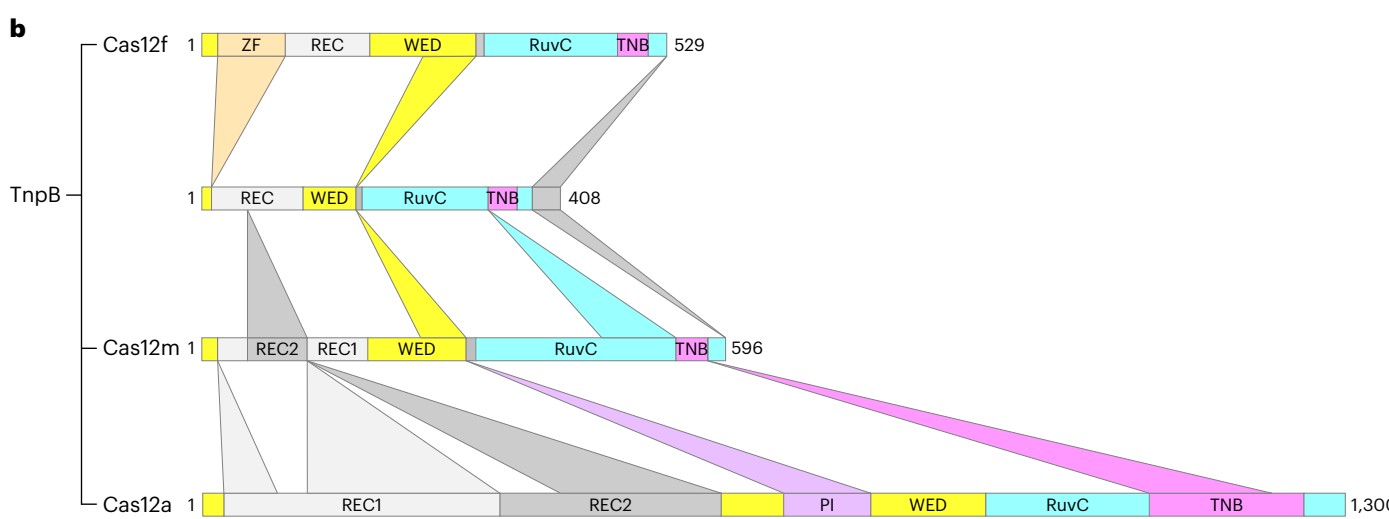

**Fig. 5 | Model of evolution from TnpB to Cas12 enzymes. a**, Structural comparison of Cas12m2 with TnpB (from *D. radiodurans* ISDra2) (PDB 8H1J), Cas12f (from an uncultured archaeon) (PDB 7C7L) and Cas12a (from *Francisella novicida*) (PDB 6I1K). Second Cas12f molecule (mol. 2) and the specific REC2 insertions in Cas12m2 and Cas12a, which are responsible for recognition of the PAM-distal region, are highlighted in blue. The insertions from Cas12m2 to Cas12a are highlighted in orange. The blue cycle represents the region where TnpB acquired additional elements to engage in the adaptive immune system, and the orange cycles represent the region where domain insertions and enlargements occurred for more efficient target recognition and cleavage. **b**, Schematic of domain insertions and enlargements of Cas12 enzymes from TnpB. ZF, zinc finger domain.

The structure of Cas12m2 strongly resembles that of the minimal TnpB containing WED, REC1 (REC domain in TnpB), RuvC and TNB domains[25,26] (Fig. 4a). Both enzymes adopt bilobed architecture and accommodate the guide RNA–target DNA heteroduplex in similar manners (Fig. 4a). The PAM duplex (transposon-associated motif (TAM) duplex in TnpB) is commonly bound between the WED and REC1 domains, and the guide RNA scaffold is accommodated within the groove formed by the WED and RuvC domains. These structural similarities confirm that the RNA-guided DNA-targeting mechanism is highly conserved between TnpB and Cas12m2.

In contrast to these conserved structural features, there are also important structural differences. TnpB only recognizes a 12-bp guide RNA–target DNA heteroduplex, whereas Cas12m2 recognizes a longer 17-bp heteroduplex (Fig. 4b). This difference is mainly due

to the characteristic REC2 insertion into the REC1 domain in Cas12m2 (Fig. 4a,b). TnpB lacks this REC2 domain, and thus the end of the guide RNA–target DNA heteroduplex is not recognized and the heteroduplexes are disordered beyond the 12th base pair. Therefore, these structural findings suggest that Cas12m2 acquired the ability to recognize the PAM-distal region of the guide RNA–target DNA heteroduplex through REC2 insertion into the core structure of TnpB. These features contributed to evolution toward CRISPR–Cas adaptive immunity.

## Discussion

In this study, we determined the cryo-EM structure of the Cas12m2–crRNA–target DNA ternary complex, revealing that the non-canonical HDD-type RuvC catalytic triad forms an NTS-binding site via coordination of a single magnesium ion. Intriguingly, when we introduced counter-mutations to convert the HDD motif of Cas12m2 into a canonical DED motif, DNA-cleavage activity was not restored (data not shown). This observation indicates that there could be more reasons why Cas12m2 lost DNA-cleavage activity aside from the HDD-type RuvC triad. In a recent study, we divided the Cas12m family enzymes further into four clades, based on the conserved RuvC catalytic triad[11] (clade 1, HEK, HQK; clade 2, HDD; clade 3, DED, NED; clade 4, DED, DND, NED). Because the other Cas12m enzyme clades have different catalytic residues, they could interfere with invading MGEs through mechanisms distinct from that of Cas12m2. Notably, clade 1 (HQK) may directly bind the NTS within the positively charged RuvC active site, while clade 3 and clade 4 may facilitate two-divalent cation-dependent DNA hydrolysis within the canonical DED-type RuvC active site. Biochemical and structural characterizations of the other Cas12m enzyme clades are required to further clarify the molecular mechanisms of Cas12m family enzymes.

Most Cas12 enzymes auto-process their own pre-crRNA into mature crRNA without the involvement of a host ribonuclease. Cas12a and Cas12i process their pre-crRNA at their WED domain by metal-independent, acid–base catalytic mechanisms[21,24,27,28], whereas Cas12c and Cas12j process their pre-crRNA at the RuvC catalytic site by a metal-dependent mechanism[8,20]. A recent study showed that, similar to Cas12a and Cas12i, Cas12m2 cleaves its pre-crRNA upstream of the crRNA scaffold, and two conserved residues (His269 and Arg270) in the WED domain may participate in crRNA maturation[11]. However, unlike Cas12a and Cas12i, in which the 5′ end of the crRNA is surrounded by the WED domain, the 5′ end of Cas12m2–crRNA is projected away from the WED domain and exposed to the solvent in the present structure (Extended Data Fig. 8). His269 and Arg270 in the WED domain form base-specific interactions with the crRNA scaffold, suggesting that these residues are responsible for crRNA recognition, rather than crRNA processing (Fig. 2d). Indeed, a processing analysis of Cas12m2 revealed that alanine substitutions of His269 and Arg270 in the WED domain and Asp485 in the RuvC catalytic site had little or no effect on the pre-crRNA-processing pattern (Extended Data Fig. 9a). These structural and biochemical analyses suggest that Cas12m2 may process its pre-crRNA in a unique manner. Intriguingly, when we analyzed the cryo-EM structure of Cas12m2 complexed with a 56-nucleotide crRNA, we observed an additional density close to the 5′ region of the crRNA (Table 1, Extended Data Fig. 9b–e and Supplementary Table 1). This density aligns well with part of the distinctive sickle-like architecture of the REC1 and REC2 domains in Cas12m2, suggesting that Cas12m2 may facilitate pre-crRNA processing through the formation of a transient asymmetric homodimer. However, because the present resolution is not sufficient to unambiguously elucidate the residues responsible for crRNA maturation, further studies are required to fully understand the pre-crRNA-processing mechanism of Cas12m2.

A structural comparison of TnpB, the putative ancestor of Cas12 enzymes, with Cas12m2, Cas12f[14] and Cas12a[19] provides insights into the evolutionary path of Cas12 enzymes (Fig. 5a,b). Cas12m2 and Cas12a commonly have REC2 insertions that recognize the PAM-distal region

of the guide RNA–target DNA heteroduplex, whereas Cas12f retains a TnpB-like REC domain. Nevertheless, Cas12f functions as an asymmetric homodimer to compensate for recognition of the PAM-distal region of the heteroduplex, with the second molecule serving as a replacement for the REC2 domain[14,15]. These observations suggest that CRISPR–Cas12 enzymes acquired the ability to recognize the PAM-distal region of the guide RNA–target DNA heteroduplex to engage in CRISPR–Cas adaptive immunity. All Cas12 enzymes except for Cas12f and Cas12k (which are both associated with one or more additional protein molecules) have the REC2 insertion to recognize the PAM-distal region[14,15,29,30], supporting our suggestion. Structural comparison between Cas12m2 and Cas12a provides insights into the maturation of Cas12 enzymes (Fig. 5a,b). Notably, Cas12a has larger REC domains and a TNB domain[19], which are responsible for target DNA recognition and loading, respectively. In addition, Cas12a contains a PAM interacting (PI) domain, which extensively interacts with the PAM duplex[19] (Fig. 5a,b). These domain enlargements and the insertion may have been acquired to target the dsDNA of invading phage more efficiently and/or to compete with anti-CRISPR systems.

## Online content

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

## Methods

### Protein and RNA preparation for structural analysis

The Cas12m2 protein for structural analysis was expressed and purified using the protocol reported previously[14,20]. Briefly, the N-terminally His$_6$-tagged Cas12m2 protein was expressed in *E. coli* Rosetta 2 (DE3). *E. coli* cells were cultured at 37 °C until the OD$_{600}$ reached 0.8, and protein expression was then induced by the addition of 0.1 mM isopropyl β-D-thiogalactopyranoside (Nacalai Tesque). *E. coli* cells were further cultured at 20 °C overnight and collected by centrifugation. The cells were then resuspended in buffer A (20 mM HEPES–NaOH, pH 7.6, 20 mM imidazole and 1 M NaCl), lysed by sonication and centrifuged. The supernatant was mixed with 3 ml Ni-NTA Superflow resin (Qiagen), and the mixture was loaded into an Econo-Column (Bio-Rad). The protein was eluted with buffer B (20 mM HEPES–NaOH, pH 7.6, 0.3 M imidazole, 0.3 M NaCl) and then loaded onto a 5-ml HiTrap SP HP column (GE Healthcare) equilibrated with buffer C (20 mM HEPES–NaOH, pH 7.6, and 0.3 M NaCl). The protein was eluted with a linear gradient of 0.3–2 M NaCl and further purified by chromatography on a Superdex 200 column (GE Healthcare) equilibrated in buffer D (20 mM HEPES–NaOH, pH 7.6, 0.5 M NaCl). The purified proteins were stored at −80 °C until use. The crRNA was transcribed in vitro with T7 RNA polymerase and purified by 10% denaturing (7 M urea) polyacrylamide gel electrophoresis.

### Electron microscopy sample preparation and data collection

The Cas12m2–crRNA–target DNA complex was reconstituted by mixing purified Cas12m2, the 56-nucleotide crRNA, the 36-nucleotide target DNA and the 36-nucleotide non-target DNA at a molar ratio of 1:1.2:1.5:1.5. The Cas12m2–crRNA binary complex was reconstituted by mixing purified Cas12m2 and the 56-nucleotide crRNA at a molar ratio of 1:1.2. The ternary and binary complexes were purified by size-exclusion chromatography on a Superdex 200 Increase 10/300 column (GE Healthcare) equilibrated with buffer E (20 mM HEPES–NaOH, pH 7.6, 50 mM NaCl, 2 mM MgCl$_2$ and 10 μM ZnCl$_2$). The purified complex solution (A$_{260}$ = 4) was then applied to Au 300-mesh R1.2/1.3 grids (Quantifoil), and, after adding 3 μl amylamine, they were glow discharged in a Vitrobot Mark IV system (FEI) at 4 °C with a waiting time of 10 s and a blotting time of 4 s under 100% humidity conditions. The grids were plunge frozen in liquid ethane and cooled to the temperature of liquid nitrogen.

Micrographs for all datasets were collected on a Titan Krios G3i microscope (Thermo Fisher Scientific) running at 300 kV and equipped with a Gatan Quantum LS Energy Filter (GIF) and a Gatan K3 Summit direct electron detector in electron-counting mode (University of Tokyo, Japan). Movies were recorded at a nominal magnification of 105,000×, corresponding to a calibrated pixel size of 0.83 Å, with a total dose of approximately 50 electrons per Å$^2$ per 48 frames using EPU software (Thermo Fisher Scientific). The dose-fractionated movies were subjected to beam-induced motion correction and dose weighting using MotionCor2 (ref. 31) implemented in RELION-3.1 (ref. 32), and contrast transfer function (CTF) parameters were estimated using patch-based CTF estimation in cryoSPARC version 3.3.2 (ref. 33).

### Single-particle cryo-EM data processing

Data were processed using cryoSPARC[33]. For the ternary complex, 5,067,096 particles were initially selected from the 3,570 motion-corrected and dose-weighted micrographs using a two-dimensional (2D) reference and extracted at a pixel size of 3.32 Å. These particles were subjected to several rounds of 2D classification to curate particle sets. The particles were further curated by heterogeneous refinement, using a map derived from ab initio reconstruction as the template. The selected 1,030,197 particles were then re-extracted at a pixel size of 1.16 Å and subjected to 3D variability analysis[34]. The resulting maps with different conformations were used for subsequent heterogeneous refinement. The selected particles after heterogeneous refinement

were refined using non-uniform refinement with optimization of the CTF value[35], yielding maps at resolutions of 2.87 Å (state I) and 3.08 Å (state II), according to the FSC criterion of 0.143 (ref. 36). The local resolution was estimated with BlocRes in cryoSPARC.

For the binary complex, 6,034,518 particles were initially selected from a total of 8,004 motion-corrected and dose-weighted micrographs using a 2D reference and extracted at a pixel size of 3.32 Å. The particles were further curated by heterogeneous refinement using a map derived from ab initio reconstruction as the template. The selected 215,565 particles were then re-extracted at a pixel size of 1.66 Å and subjected to 3D classification without alignment, using a mask focused on the second subunit. After 3D classification, the selected particles were refined using non-uniform refinement, yielding a map at a resolution of 3.73 Å, according to the FSC criterion of 0.143. The local resolution was estimated with BlocRes in cryoSPARC.

### Model building and validation

The model was built using the predicted model of the Cas12m2 protein created by AlphaFold2 (the AlphaFold model) as the reference, followed by manual model building with Coot[37,38]. The model was refined using phenix.real_space_refine version 1.20.1 (ref. 39) with secondary structure and metal coordination restraints. The metal coordination restraints were generated using ReadySet, as implemented in PHENIX. Structure validation was performed using MolProbity[40]. Residues 592–596 and nucleotides −36 to −30 of the crRNA, nucleotides −9 to −5 of the TS and nucleotides 22–26 of the NTS were not included in the final full R-loop state model because these regions were not well resolved on the density map. Residues 53–120, 499–506 and 592–596 and nucleotides −36 to −30 and 13–20 of the crRNA, nucleotides −9 to 5 and 27 of the TS and nucleotides −10 and 13–26 of the NTS were not included in the final intermediate state model. In the binary complex, residues 87–90, 488–508 and 592–596 of Cas12m2.1, residues 1–17, 65–94, 189–250 and 262–596 of Cas12m2.2 and nucleotides −36 to −30 and 16–20 of the crRNA were not included in the final model. The cryo-EM density maps were calculated with UCSF ChimeraX[41], and the molecular graphics shown in the figures were prepared with CueMol (http://www.cuemol.org).

### Assembly of Cas12m2 PAM and REC mutant plasmids

Plasmids used for protein expression of Cas12m2 PAM and REC mutants were assembled by PCR mutagenesis (Q5 polymerase, NEB; T4 DNA ligase, NEB) using the primers listed in Supplementary Table 2, with pML-1B-Cas12m as the PCR template[11]. Sequence-verified plasmids were used as templates for amplification of their inserts for restriction ligation assembly of pCas-Cas12m2-[x] plasmids, which were used for in vivo transcriptional silencing assays, with pCas-Cas12m2 as the template for backbone amplification (Supplementary Table 2).

### In vivo transcriptional silencing assay

The transcriptional silencing activity of Cas12m variants in vivo was assessed using the protocol reported previously[11], with minor modifications. Briefly, *E. coli* cells harboring pTarget-operon and either pCRISPR-A1, pCRISPR-E1 or pCRISPR-NT were made chemically competent and transformed with either pCas-Cas12m, pCas-dCas12m or pCas-Cas12m-[x] (the ten alanine-substitution mutants). After recovery, 5 μl of transformation mix was diluted with 195 μl M9TG medium in a 96-well 2-ml masterblock (Greiner) in triplicate, sealed with a gas-permeable membrane and grown at 37 °C and 750 r.p.m. for 20 h. Cells were diluted 1:1,000 in fresh M9TG medium in a total volume of 200 μl in a 96-well masterblock and grown overnight at 37 °C. Overnight cultures were diluted 1:10 in 200 μl PBS and analyzed using a Synergy Mx microplate reader (BioTek) and Gen5 (software version 3.10.06). The fluorescence of GFP (excitation, 488 nm; emission, 512 nm) and RFP (excitation, 532 nm; emission, 588 nm) was measured with gains of 91 and 140, respectively. Fluorescence values were normalized to

blank-corrected $OD_{600}$ values, and technical triplicates were averaged. In Fig. 3h, the averages of two biological replicates are shown relative to the average normalized fluorescence of wells containing *E. coli* expressing a non-targeting spacer (pCRISPR-NT). Statistical analysis was performed using Tukey's post hoc test in case of significant two-way ANOVA results.

**Cas12m2 purification for surface plasmon resonance analyses**
The pML-1B-Cas12m2, pML-1B-dCas12m2 and pML-1B-Cas12m2-[x] plasmids were transformed into chemically competent *E. coli* Rosetta 2 (DE3) cells and purified using a modified version of the previously reported protocol[11]. Briefly, the transformed *E. coli* strains were inoculated into 800 ml LB medium supplemented with chloramphenicol and kanamycin, cultured overnight and grown to an $OD_{600}$ of 0.6–0.8 at 37 °C and 150 r.p.m. The cultures were cooled on ice for 1 h before the addition of 0.4 mM isopropyl 1-thio-β-D-galactopyranoside and then grown for 16–20 h at 20 °C and 120 r.p.m. Cultured cells were pelleted, washed and resuspended in lysis buffer (1 M NaCl, 20 mM imidazole, 20 mM HEPES, pH 7.6) freshly supplemented with protease inhibitors (Roche cOmplete, EDTA-free) and lysed by sonication (Bandelin Sonopuls). The lysate was cleared (30 min, 30,000$g$), and the supernatant was filtered (0.45 μm). The filtrate was applied to a nickel column (HisTrap HP, GE Life Sciences), which was washed (500 mM NaCl, 20 mM imidazole, 20 mM HEPES, pH 7.6) and eluted (500 mM NaCl, 300 mM imidazole, 20 mM HEPES, pH 7.6) using an ÄKTA FPLC system. Elution fractions containing the expected band were pooled, concentrated in SEC buffer (500 mM NaCl, 20 mM HEPES, pH 7.6, 1 mM DTT) through centrifugal filters (Amicon Ultra-15, MW cutoff 30,000 Da) and further purified by size-exclusion chromatography (HiLoad 16/600 Superdex 200) by FPLC. Pure fractions were pooled and snap frozen in liquid nitrogen until analysis by surface plasmon resonance (SPR).

**Surface plasmon resonance**
SPR spectroscopy was performed at 25 °C on a Biacore T100 system (Cytiva). A series S CM5 sensor chip surface was modified with 2,500 response units of streptavidin (Invitrogen) using an amine coupling kit (Cytiva). Flow cell 1 was used as the reference. A 50-nucleotide biotinylated oligonucleotide containing a DNA target site was annealed to its complementary strand (Supplementary Table 2; IDT) in 25 mM HEPES, pH 7.5, containing 150 mM KCl, and 20 response units were immobilized on flow cell 2. Ribonucleotide protein complexes were formed by mixing purified protein variants with a twofold molar excess of PAM-SCNR RNA or a non-target RNA (Supplementary Table 2) to a final concentration of 500 nM protein in SPR buffer (20 mM HEPES, pH 7.8, 150 mM KCl, 10% glycerol, 5 mM $MgCl_2$, 1 mM DTT and 0.05% Tween-20). Dilutions of this complex were injected over the chip surface at 50 μl min$^{-1}$. Flow cells were regenerated using three consecutive injections of 4 M $MgCl_2$, followed by an injection of the complementary oligonucleotide (1 μM) for 100 s at 5 μl min$^{-1}$ in flow cell 2. Wild-type Cas12m2 displays multiphasic association and dissociation kinetics, which is most prominent at protein concentrations above 100 nM[11]; therefore, sensorgrams at lower concentrations (8–63 nM) were fitted to a two-state model using BIAevaluation software (Cytiva) to obtain a $K_D$ estimate. After double-reference correction, the remaining bulk effects masked the severely reduced binding capabilities of the mutant proteins in the association phase in the sensorgrams; therefore, the binding of Cas12m2 variants at 500 nM to the target site was quantified from the stable binding levels at the start of the dissociation phase (average signal of a 10-s window, 20 s after dissociation) relative to that of wild-type Cas12m2.

**Pre-crRNA processing assay**
Pre-crRNA processing was achieved using an in vitro transcription and translation system, as described previously[11]. A pCas and pCRISPR plasmid mixture expressing Cas12m2 and a CRISPR array (containing four

spacers) was incubated at 29 °C for 5 h in a thermocycler. Total RNA was then extracted using a Direct-zol RNA MiniPrep kit, according to the manufacturer's instructions (Zymo Research). For northern blotting analysis, 5 μg of each RNA sample obtained from cell-free transcription–translation was fractionated on an 8% polyacrylamide gel (7 M urea) at 300 V for 140 min. RNA was transferred onto Hybond-XL membranes (Amersham Hybond-XL, GE Healthcare) using an electroblotter at 50 V for 1 h at 4 °C (Tank-Elektroblotter Web M, PerfectBlue) and cross-linked with UV light at 0.12 J (UV lamp T8C, 254 nm, 8 W). Hybridization proceeded overnight in 17 ml of Roti-Hybri-Quick buffer, with 5 μl of [γ-$^{32}$P]ATP end-labeled oligodeoxyribonucleotides at 42 °C. The membrane was visualized using a Phosphorimager (Typhoon FLA 7000, GE Healthcare).

**Reporting summary**
Further information on research design is available in the Nature Portfolio Reporting Summary linked to this article.

## Data availability
The structural models have been deposited in the PDB under accession codes PDB 8HHL (full R-loop state), PDB 8HHM (intermediate state) and PDB 8HIO (binary complex). The cryo-EM density maps have been deposited in the EMDB under accession codes EMD-34803 (full R-loop state), EMD-34804 (intermediate state) and EMD-34824 (binary complex). Source data are provided with this paper.

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

## Acknowledgements

We thank C. Liao, A. Migur and C.L. Beisel (HZI, Würzburg, Germany) for contributing to the guide-processing analysis. J.H.G.L. and C.L. are supported by the gravitation program CancerGenomiCs.nl from the Netherlands Organisation for Scientific Research, part of the Oncode

Institute, which is partly financed by the Dutch Cancer Society. O.N. was supported by AMED (grant numbers JP19am0401005 and JP21km0908001), the Platform Project for Supporting Drug Discovery and Life Science Research (Basis for Supporting Innovative Drug Discovery and Life Science Research) from AMED under grant number JP22am0101115 and the Cabinet Office, Government of Japan, Public/Private R&D Investment Strategic Expansion Program (grant number JPJ008000).

## Author contributions

S.N.O. performed structural analyses with assistance from R.N., H.H., T.K., Y.K. and Y.I.; S.N.O., R.N., H.H. and Y.I. built the model and performed structural refinement; W.Y.W., C.S., R.V.W., C.L. and J.H.G.L. performed in vitro and in vivo biological experiments; J.v.d.O. conceived the project; S.N.O., R.N. and O.N. wrote the paper with help from all authors; O.N. supervised the research.

## Competing interests

O.N. is a cofounder, board member and scientific advisor of Curreio. Two patent applications have been filed related to this work, with J.v.d.O. and W.Y.W. as inventors. J.v.d.O. is an advisor for NTrans Technologies, Scope Biosciences and Hudson River Biotechnology. The other authors declare no competing interests.

## Additional information

**Extended data** is available for this paper at https://doi.org/10.1038/s41594-023-01042-3.

**Correspondence and requests for materials** should be addressed to John van der Oost or Osamu Nureki.

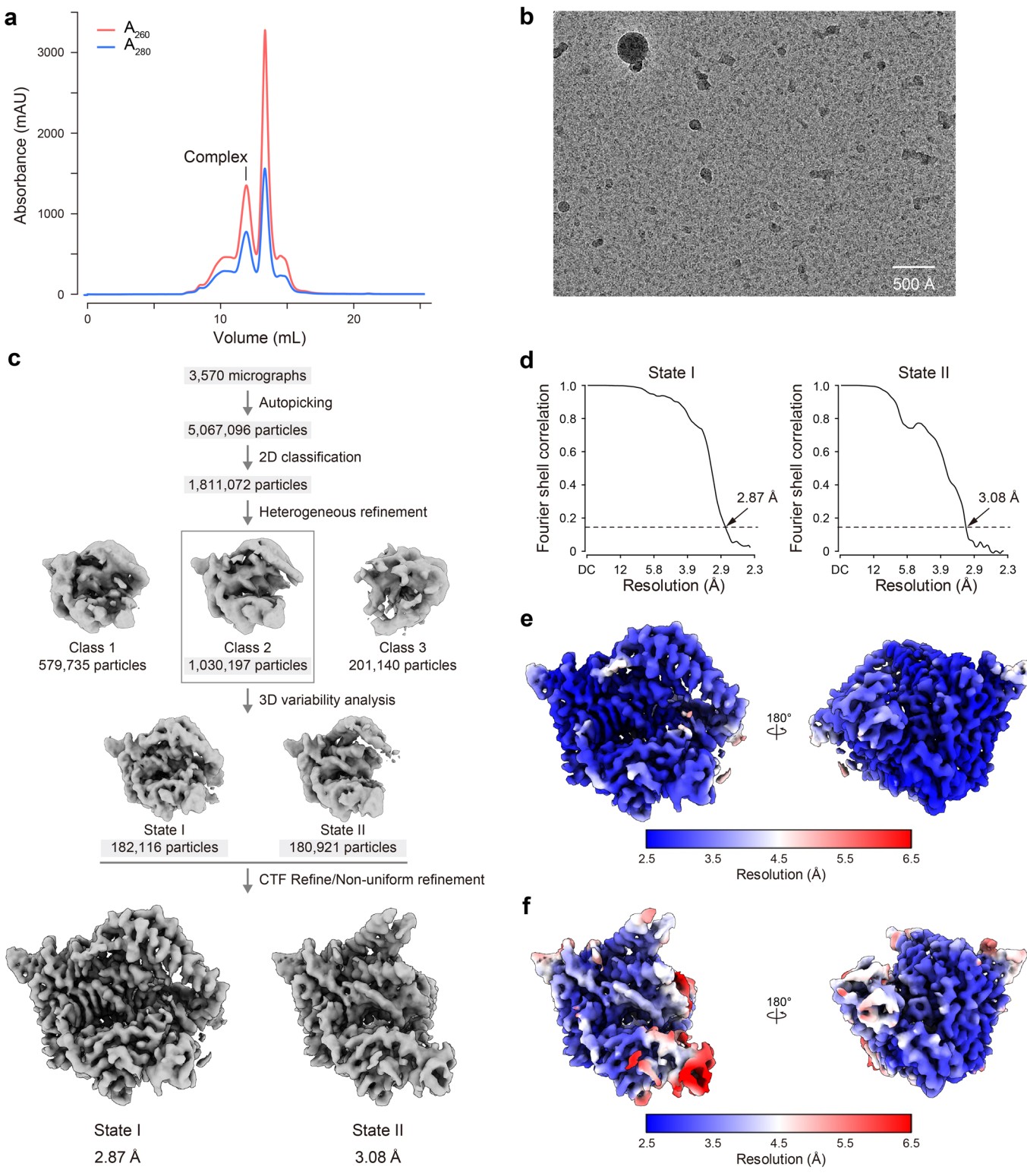

**Extended Data Fig. 1 | Single-particle cryo-EM analysis of the Cas12m2–crRNA–target DNA ternary complex.** (**a**) Size-exclusion chromatography profile of the Cas12m2–crRNA–target DNA ternary complex. The peak fraction was used for cryo-EM analysis. (**b**) A representative cryo-EM image of the Cas12m2–crRNA–target DNA complex, recorded on a 300-kV Titan Krios microscope with a K3 camera. (**c**) Single-particle cryo-EM image processing workflow. (**d**) Fourier shell correlation (FSC) curve for the 3D reconstruction, with the gold-standard cut-off (FSC = 0.143) marked with a black dotted line. (**e and f**) Local-resolution cryo-EM density maps of State I (full R-loop state) (**e**) and State II (intermediate state) (**f**).

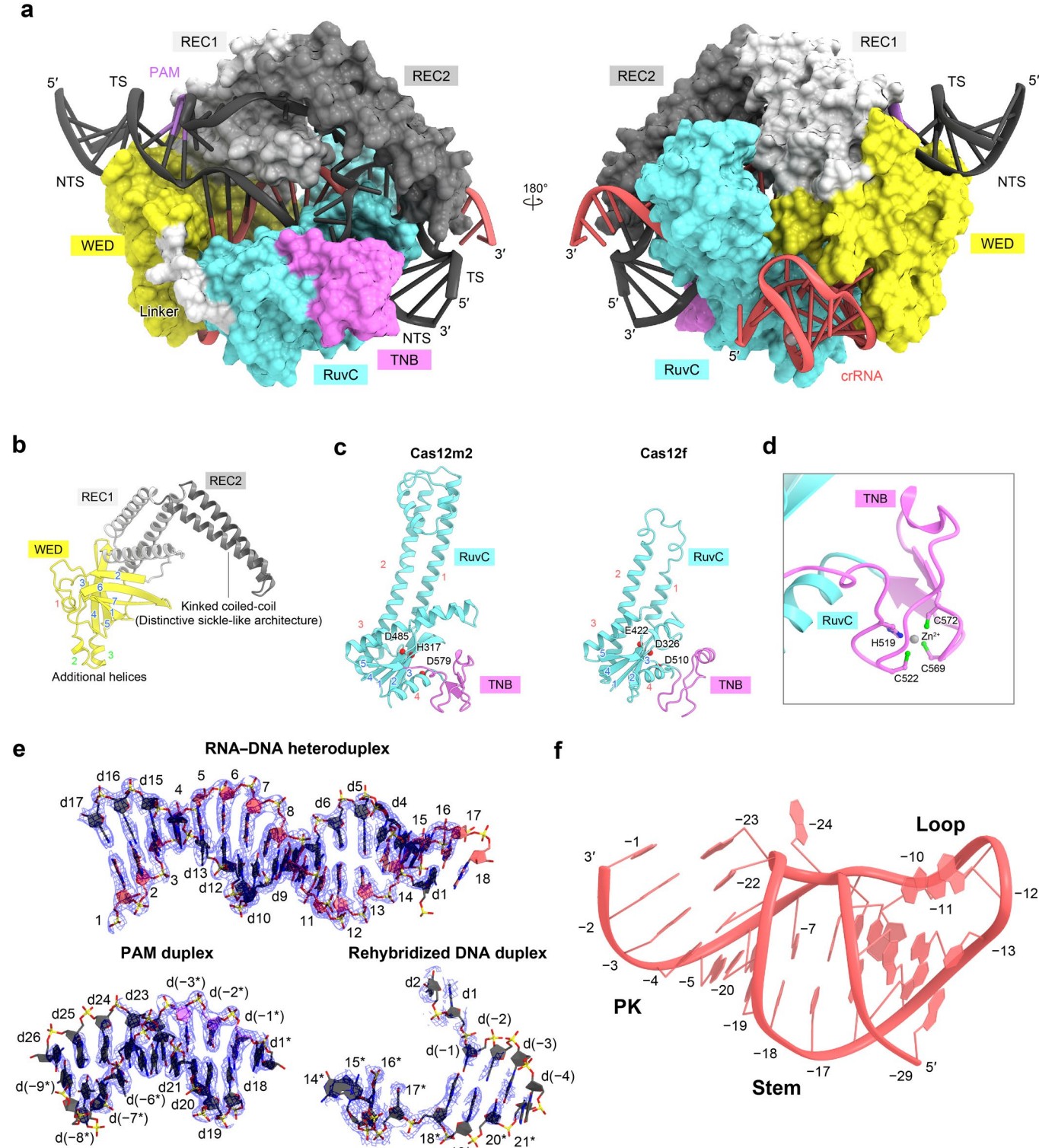

**Extended Data Fig. 2 | Overall, domain, and nucleic acid structures of the Cas12m2–crRNA–target DNA ternary complex.** (**a**) Surface representation of the Cas12m2–crRNA–target DNA ternary complex. (**b**) The REC lobe structure of Cas12m2. The WED domain comprises an oligonucleotide/oligosaccharide-binding fold (the conserved β strands (blue) and α-helix (red) are numbered). The two additional α-helices are identified with green numbers. The REC1 and REC2 domains form a distinctive sickle-like architecture. (**c**) The NUC lobe

structure of Cas12m2 and that of Cas12f (from an uncultured archaeon) (PDB ID: 7C7L). The NUC lobes consist of the RuvC and TNB domains. The RuvC domains comprise the RNase H fold (the conserved β strands (blue) and α-helices (red) are numbered). (**d**) Zinc binding site in the TNB domain. (**e**) The cryo-EM densities for the RNA–DNA heteroduplex, PAM duplex, and rehybridized DNA duplex are shown as a blue mesh. (**f**) Structure of the crRNA scaffold.

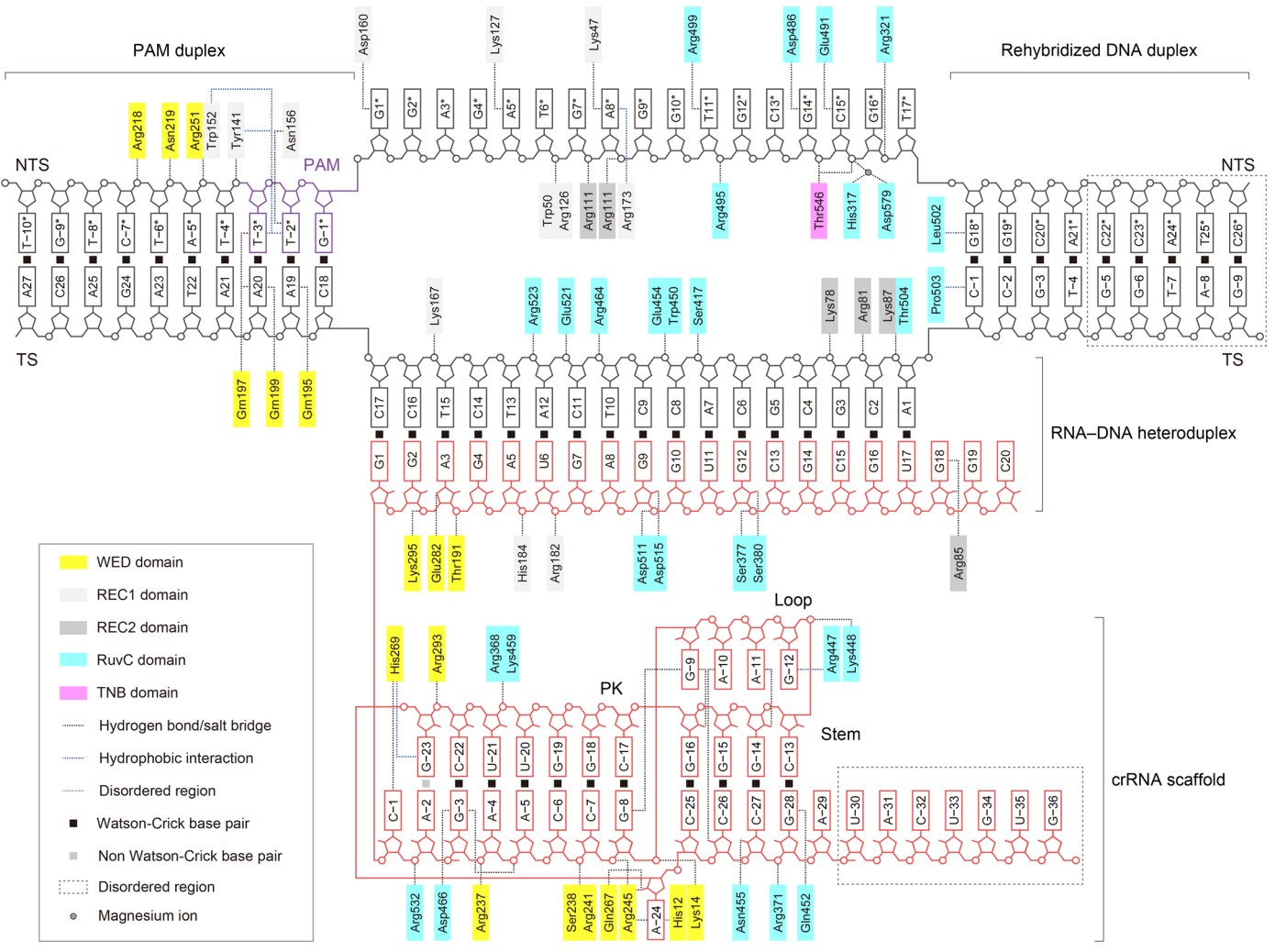

**Extended Data Fig. 3 | Nucleic acid recognition and *in vitro* DNA binding profile.** Schematic of nucleic acid recognition by Cas12m2. The residues that interact with nucleic acids through their main chains are shown in parentheses. The disordered region is enclosed in a dashed box.

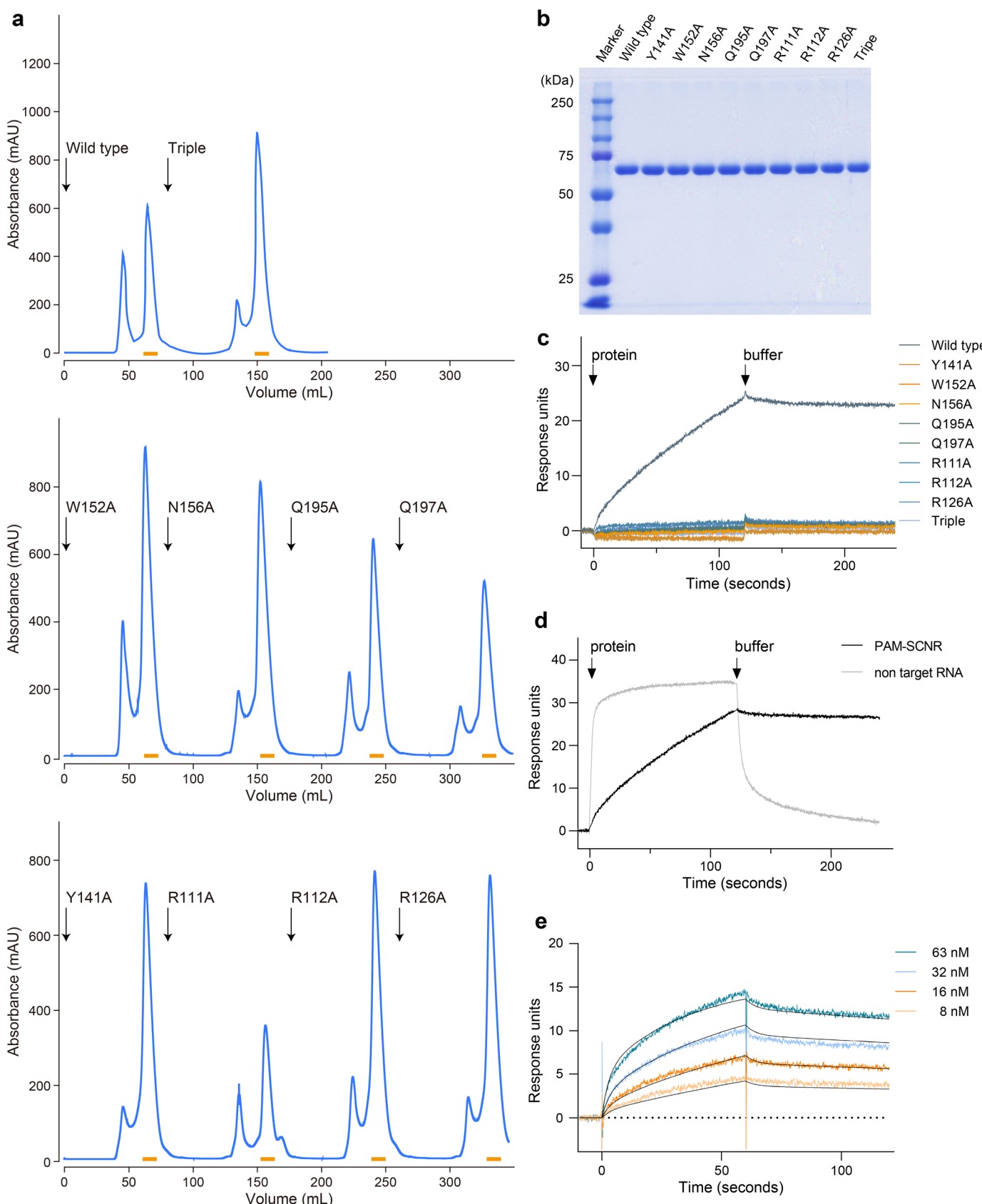

**Extended Data Fig. 4 | See next page for caption.**

**Extended Data Fig. 4 | *In vitro* surface plasmon resonance analysis. (a)** SEC chromatograms of the final purification step of the Cas12m2 proteins analyzed in this study. Arrows indicate the loading of each protein onto the column. Elution fractions that were pooled for protein analysis are highlighted in orange. Triple: R111A/R112A/R126A triple mutant. (**b**) SDS-PAGE analysis of SEC-purified proteins. Two micrograms of protein were loaded per well. The theoretical molecular mass of the variants is ~68 kDa. (**c**) SPR sensorgrams of 500 nM wild type Cas12m2 and variants bound to PAM-SCNR RNA, binding to a 50-base pair dsDNA oligonucleotide containing a target site. (**d**) SPR sensorgram of 500 nM wild type Cas12m2 binding to the DNA target site with PAM-SCNR RNA (black) and non-target RNA (gray). With the non-target RNA, Cas12m2 rapidly binds but is unable to form a stable complex and dissociates quickly. With the target RNA, a stable complex is formed. (**e**) Fit of a two-state binding model (black lines) to SPR sensorgrams of wild type Cas12m with PAM-SCNR RNA binding to the target site (colored curves) returns an estimated $K_D$ of 2.7 nM.

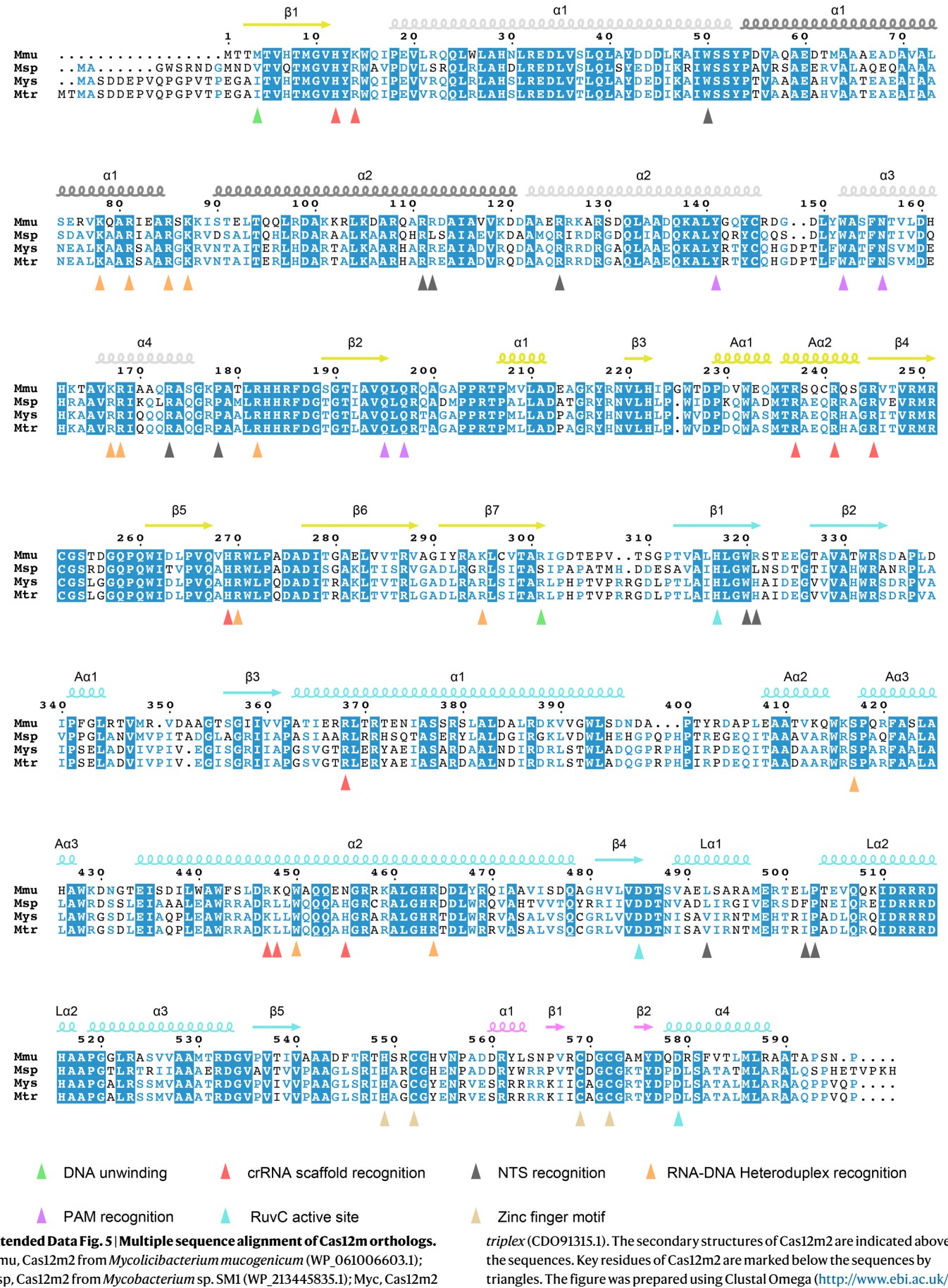

**Extended Data Fig. 5 | Multiple sequence alignment of Cas12m orthologs.**
Mmu, Cas12m2 from *Mycolicibacterium mucogenicum* (WP_061006603.1);
Msp, Cas12m2 from *Mycobacterium* sp. SM1 (WP_213445835.1); Myc, Cas12m2
from *Mycobacterium* (WP_036473531.1); Mtr, Cas12m2 from *Mycobacterium*
*triplex* (CDO91315.1). The secondary structures of Cas12m2 are indicated above
the sequences. Key residues of Cas12m2 are marked below the sequences by
triangles. The figure was prepared using Clustal Omega (http://www.ebi.ac.uk/
Tools/msa/clustalo) and ESPript3 (http://espript.ibcp.fr/ESPript/ESPript).

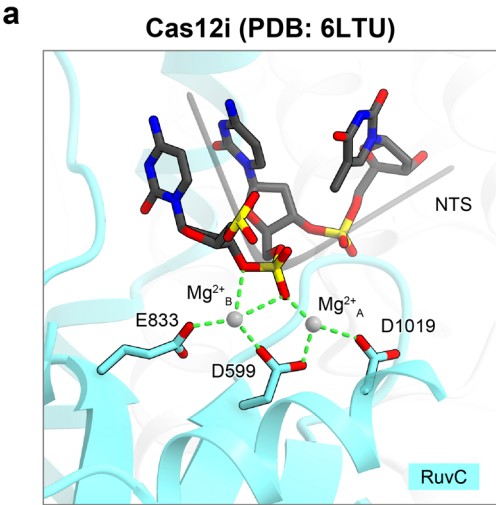

**a** Cas12i (PDB: 6LTU)

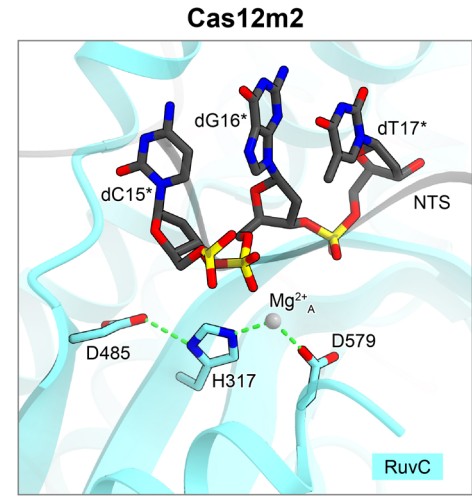

**c** Cas12m2

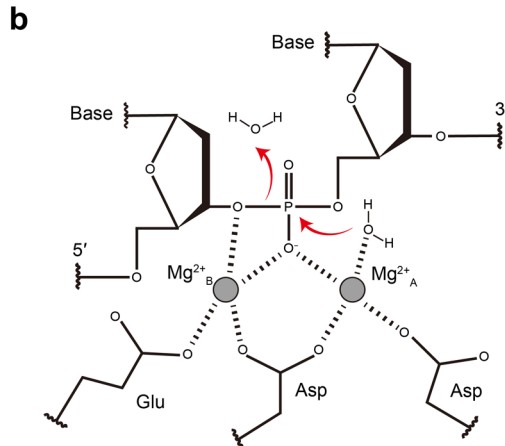

**b**

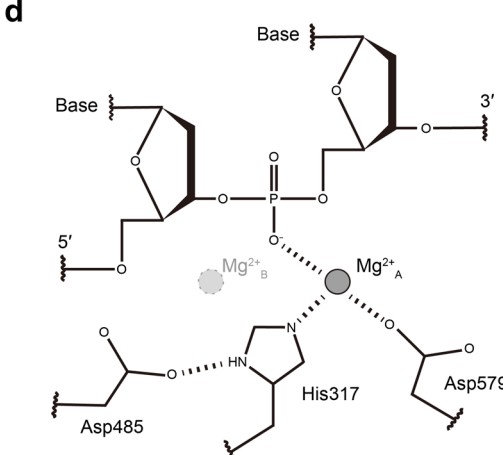

**d**

**Extended Data Fig. 6 | Non-canonical RuvC active site.** (**a and c**) Structures of the RuvC active site of Cas12i2 (**a**) (Cas12i2 from an unidentified source) (PDB ID: 6LTU) and Cas12m2 (**c**). While Cas12i2 coordinates two magnesium ions on either side of the scissile phosphate, Cas12m2 coordinates only one magnesium ion, which corresponds to magnesium ion A in Cas12i2. (**b and d**) Schematics of the RuvC active site representing the canonical DED-type (**b**) and non-canonical HDD-type (**d**). In the canonical DED-type RuvC active site, the first magnesium ion (A) activates the nucleophilic water molecule, while the second magnesium ion (B) stabilizes the transition state, resulting in two divalent cation-dependent DNA hydrolysis (**b**). In the non-canonical HDD-type RuvC active site, no density corresponding to magnesium ion B was observed, resulting in the loss of cleavage activity for the target DNA of Cas12m2 (**d**).

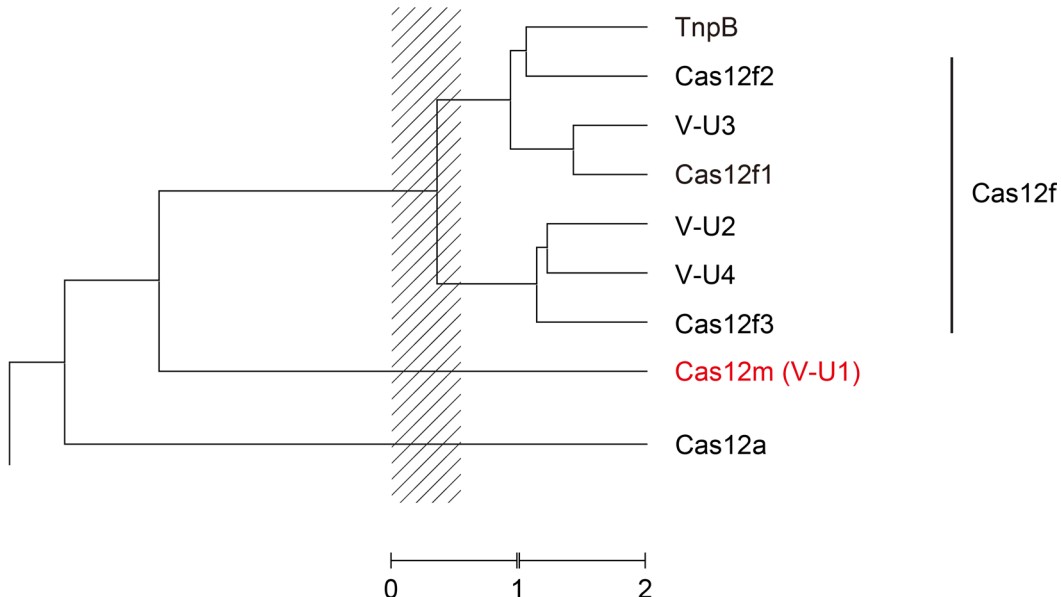

**Extended Data Fig. 7 | UPGMA dendrogram showing the similarity between different families of type V effectors.** The dendrogram was built using the UPGMA (unweighted pair group method with arithmetic mean) method and is based on HHalign matrix[42] scores calculated for all families against all pairwise alignments with a length coverage of 33% or more. The alignments for the respective families were taken from previous report[2], except for the Cas12m family, for which an updated alignment (104 proteins) was used. The dashed rectangle corresponds to a tree depth (D) between 1.5 and 2 (D = 2 roughly corresponds to the pairwise HHsearch similarity score of exp(2D) ≈ 0.02 relative to the self-score) and reflects the D where the subtype assignment is uncertain and requires additional consideration.

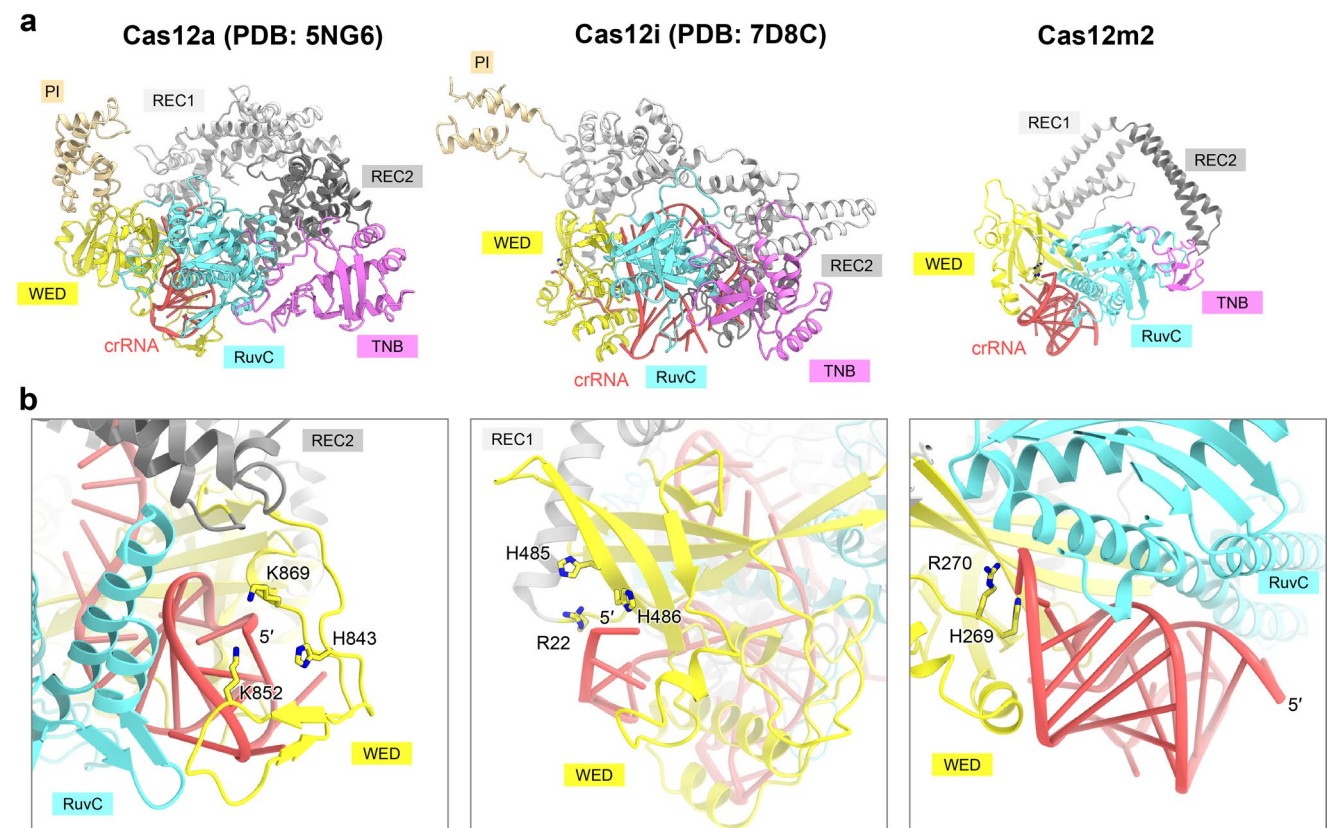

**Extended Data Fig. 8 | Pre-crRNA processing mechanism of Cas12a, Cas12i, and Cas12m2.** Overall structures (**a**) and close-up views of the 5′ end of crRNA (**b**). Cas12a (from *Francisella novicida*) (PDB ID: 5NG6) and Cas12i (from *Lachnospiraceae bacterium ND2006*) (PDB ID: 7D8C) process their pre-crRNAs at the WED domain, and the 5′ end of the crRNA is surrounded by the WED domain.

In contrast, the 5′ end of the Cas12m2 crRNA is projected away from the WED domain and exposed to the solvent. Target DNA and spacer region of crRNA are omitted for clarity in the Cas12m2–crRNA–target DNA ternary complex structure.

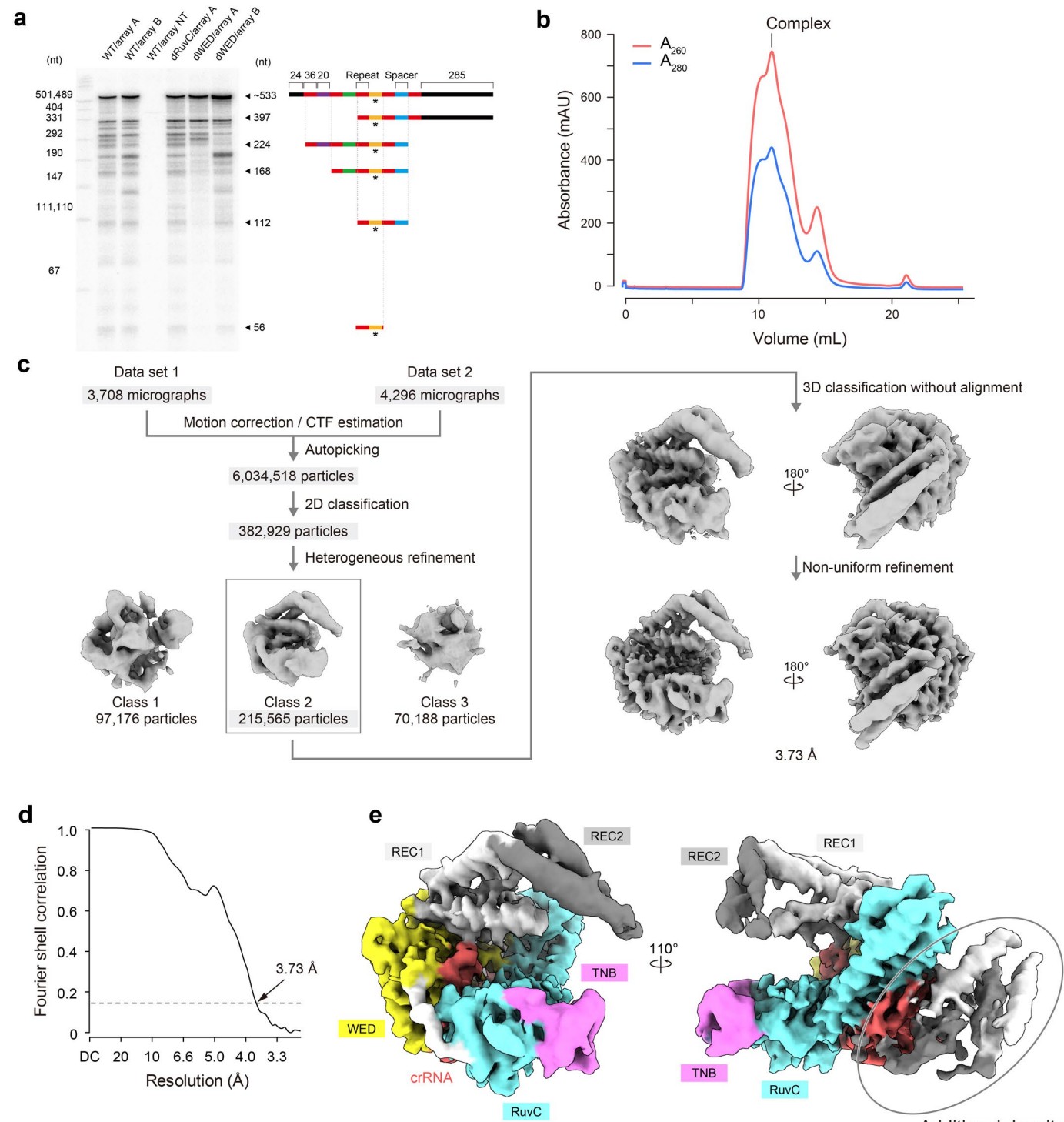

**Extended Data Fig. 9 | Single-particle cryo-EM analysis of the Cas12m2–crRNA binary complex.** (**a**) Pre-crRNA processing activities of wild type (WT), H269A/R270A (dWED), and D485A (dRuvC) in TXTL. A plasmid expressing the Cas12m2 CRISPR array containing four spacers was incubated with a plasmid expressing WT, dWED, or dRuvC Cas12m2 in TXTL. The cleavage products were visualized by northern blotting, using a probe that binds to the third spacer of array A (yellow, shown as an asterisk). Array B is similar to array A; only the order

of spacers is shifted. (**b**) Size-exclusion chromatography profile of the Cas12m2–crRNA binary complex. The peak fraction was used for cryo-EM analysis. (**c**) Single-particle cryo-EM image processing workflow. (**d**) FSC curve for the 3D reconstruction, with the gold-standard cut-off (FSC = 0.143) marked with a black dotted line. (**e**) Cryo-EM density map of the Cas12m–crRNA binary complex. The additional three-helix-like density is indicated by a circle.

# Reporting Summary

## Statistics

For all statistical analyses, confirm that the following items are present in the figure legend, table legend, main text, or Methods section.

| n/a | Confirmed | |
|---|---|---|
| ☐ | ☒ | The exact sample size (*n*) for each experimental group/condition, given as a discrete number and unit of measurement |
| ☐ | ☒ | A statement on whether measurements were taken from distinct samples or whether the same sample was measured repeatedly |
| ☒ | ☐ | The statistical test(s) used AND whether they are one- or two-sided *Only common tests should be described solely by name; describe more complex techniques in the Methods section.* |
| ☒ | ☐ | A description of all covariates tested |
| ☐ | ☒ | A description of any assumptions or corrections, such as tests of normality and adjustment for multiple comparisons |
| ☐ | ☒ | A full description of the statistical parameters including central tendency (e.g. means) or other basic estimates (e.g. regression coefficient) AND variation (e.g. standard deviation) or associated estimates of uncertainty (e.g. confidence intervals) |
| ☐ | ☒ | For null hypothesis testing, the test statistic (e.g. *F*, *t*, *r*) with confidence intervals, effect sizes, degrees of freedom and *P* value noted *Give P values as exact values whenever suitable.* |
| ☒ | ☐ | For Bayesian analysis, information on the choice of priors and Markov chain Monte Carlo settings |
| ☒ | ☐ | For hierarchical and complex designs, identification of the appropriate level for tests and full reporting of outcomes |
| ☒ | ☐ | Estimates of effect sizes (e.g. Cohen's *d*, Pearson's *r*), indicating how they were calculated |

*Our web collection on statistics for biologists contains articles on many of the points above.*

## Software and code

Policy information about availability of computer code

| Data collection | EPU (version 2.12), Synergy MX microplate reader, Biacore T100 |
|---|---|
| Data analysis | cryoSPARC (version 3.3.2), COOT (version 0.9), UCSF ChimeraX (version 1.1.1), CueMol2 (http://www.cuemol.org/ version 2.2.3.443), phenix (version 1.20.1), Gen5 (version 3.10.06), BiaEvaluation |

For manuscripts utilizing custom algorithms or software that are central to the research but not yet described in published literature, software must be made available to editors and reviewers. We strongly encourage code deposition in a community repository (e.g. GitHub). See the Nature Portfolio guidelines for submitting code & software for further information.

## Data

Policy information about availability of data

All manuscripts must include a data availability statement. This statement should provide the following information, where applicable:
- Accession codes, unique identifiers, or web links for publicly available datasets
- A description of any restrictions on data availability
- For clinical datasets or third party data, please ensure that the statement adheres to our policy

The cryo-EM density map and atomic coordinate have been deposited in the Electron Microscopy Data Bank. The accession code for the maps are EMD-34803 for Cas12m2–crRNA–DNA Full R-loop, EMD-34804 for Cas12m2–crRNA–DNA intermediate, and EMD-34824 for Cas12m2–crRNA. The accession code for the coordinates are 8HHL for Cas12m2–crRNA–DNA Full R-loop, 8HHM for Cas12m2–crRNA–DNA intermediate, and 8HIO for Cas12m2–crRNA.

# Human research participants

Policy information about studies involving human research participants and Sex and Gender in Research.

| | |
|---|---|
| Reporting on sex and gender | not applicable |
| Population characteristics | not applicable |
| Recruitment | not applicable |
| Ethics oversight | not applicable |

Note that full information on the approval of the study protocol must also be provided in the manuscript.

# Field-specific reporting

Please select the one below that is the best fit for your research. If you are not sure, read the appropriate sections before making your selection.

☒ Life sciences          ☐ Behavioural & social sciences          ☐ Ecological, evolutionary & environmental sciences

For a reference copy of the document with all sections, see nature.com/documents/nr-reporting-summary-flat.pdf

# Life sciences study design

All studies must disclose on these points even when the disclosure is negative.

| | |
|---|---|
| Sample size | No statistical method was used to determine the sample size. For cryo-EM analyses, sample sizes were determined by the availability of microscope time and the number of particles on electron microscopy grids enough to obtain a structure at the reported resolution. For biochemical analysis, sample size were determined based on the previous reports of this type of study and the reproducibility of results across independent experiments. |
| Data exclusions | For cryo-EM analyses, particles that did not contribute to improving map quality were excluded following the standard classification procedures in cryoSPARC. This is standard practice for structure determination by cryo-EM. For biochemical analyses, no data was excluded. |
| Replication | Biochemical experiments were performed at least three times. |
| Randomization | For cryo-EM analyses, particles were randomly assigned to half-maps for resolution determination following the standard procedures in cryoSPARC. For biochemical analyses, randomization was not performed. |
| Blinding | Blinding is not applicable to this study, since neither structural nor functional experiments included subjective assignments. |

# Reporting for specific materials, systems and methods

We require information from authors about some types of materials, experimental systems and methods used in many studies. Here, indicate whether each material, system or method listed is relevant to your study. If you are not sure if a list item applies to your research, read the appropriate section before selecting a response.

## Materials & experimental systems

| n/a | Involved in the study |
|---|---|
| ☒ ☐ | Antibodies |
| ☒ ☐ | Eukaryotic cell lines |
| ☒ ☐ | Palaeontology and archaeology |
| ☒ ☐ | Animals and other organisms |
| ☒ ☐ | Clinical data |
| ☒ ☐ | Dual use research of concern |

## Methods

| n/a | Involved in the study |
|---|---|
| ☒ ☐ | ChIP-seq |
| ☒ ☐ | Flow cytometry |
| ☒ ☐ | MRI-based neuroimaging |

