## [Peer Review File · Nature Structural & Molecular Biology]

Peer Review Information

Journal: Nature Structural and Molecular Biology

Manuscript Title: Mechanistic and evolutionary insights into a type V-M CRISPR-Cas effector enzyme

Corresponding author name(s): Osamu Nureki, John van der Oost

Editorial Notes:

Transferred manuscripts This manuscript has been previously reviewed at another journal that is not operating a transparent peer review scheme. This document only contains reviewer comments, rebuttal and decision letters for versions considered at Nature Structural & Molecular Biology.

Reviewer Comments & Decisions:

Decision Letter, initial version:
--

21st Feb 2023

Dear Dr. Nureki,

Thank you again for submitting your manuscript "Mechanistic and evolutionary insights into a type V-M CRISPR-Cas effector enzyme". I am sending you this email to primarily facilitate efficient processing your manuscript in our online system and secondarily to remind you of certain editorial requirements with respect to data deposition and presentation. For our guidance with respect to what is expected experimentally and textually in response to the reviewers, I am simply copy-pasting here for transparency and clarity reasons what I had reiterated in my email to you on February 13th.

"As also stated in our guidelines in the Nature decision letter, we have three main requests 1) fully addressing all technical concerns, including missing controls as indicated by the reviewers, 2) addressing the concerns of reviewers about the purported auto-processing mechanism of crRNA by dimerisation of Cas12m2. We would favour addressing this concern by performing structural/biochemical experiments to validate the conclusion, instead of removing or toning down the claim. 3) We request that you follow the guidance of reviewer #2 with respect to the suggested biochemical experiments and potentially the functional experiment suggested in point 6. As with every revision we expect you to issue a point by point response in the provided feedback."

We expect to see your revised manuscript within 3-6 months. If you cannot send it within this time, please contact us to discuss an extension; we would still consider your revision, provided that no similar work has been accepted for publication at NSMB or published elsewhere.

Reporting Summary:

When submitting the revised version of your manuscript, please pay close attention to our [href="https://www.nature.com/nature-portfolio/editorial-policies/image-integrity">Digital Image Integrity Guidelines.](https://www.nature.com/nature-portfolio/editorial-policies/image-integrity) and to the following points below:

Please note that all key data shown in the main figures as cropped gels or blots should be presented in uncropped form, with molecular weight markers. These data can be aggregated into a single supplementary figure item. While these data can be displayed in a relatively informal style, they must refer back to the relevant figures. These data should be submitted with the final revision, as source data, prior to acceptance, but you may want to start putting it together at this point.

SOURCE DATA: we urge authors to provide, in tabular form, the data underlying the graphical representations used in figures. This is to further increase transparency in data reporting, as detailed in this editorial (<http://www.nature.com/nsmb/journal/v22/n10/full/nsmb.3110.html>). Spreadsheets

can be submitted in excel format. Only one (1) file per figure is permitted; thus, for multi-paneled figures, the source data for each panel should be clearly labeled in the Excel file; alternately the data can be provided as multiple, clearly labeled sheets in an Excel file. When submitting files, the title field should indicate which figure the source data pertains to. We encourage our authors to provide source data at the revision stage, so that they are part of the peer-review process.

Data availability: this journal strongly supports public availability of data. All data used in accepted papers should be available via a public data repository, or alternatively, as Supplementary Information. If data can only be shared on request, please explain why in your Data Availability Statement, and also in the correspondence with your editor. Please note that for some data types, deposition in a public repository is mandatory - more information on our data deposition policies and available repositories can be found below:

<https://www.nature.com/nature-research/editorial-policies/reporting-standards#availability-of-data>

[redacted]

Sincerely,

Dimitris Typas
Associate Editor
Nature Structural & Molecular Biology
ORCID: 0000-0002-8737-1319

Author Rebuttal to Initial comments

Responses to reviewers' comments

Reviewer #1:

The authors determine structures of a Cas12 variant (V-U1), which they call Cas12m2. These proteins are smaller than most Cas12s, and provide sequence-specific defense by blocking transcription rather than cleaving the target. The latter observation is consistent with amino acid substitutions in the RuvC domain.

These structures add important detail to our understanding of Cas12 evolution and mechanism. Specifically, the structures reveal how mutations in the RuvC active site alter metal coordination and eliminate DNA cleavage, explain how the NTS is coordinated by a patch of four arginine residues, explain how REC2 engages PAM distal DNA (similar to other Cas12s), and support a nuanced understanding of the evolutionary trajectory of these proteins from TnsB to Cas12. However, claims in the abstract “that Cas12m2 processes its pre-crRNA by forming a transient, asymmetric homodimer” are not sufficiently supported by the data. A structure of the dimeric pre-CRISPR RNA processing complex would increase the novelty of this work, but data in the current manuscript do not significantly change our understanding of function or evolution of these proteins. While this assessment of impact is clearly subjective, it certainly does not undermine the science, nor does it contradict earlier statements made about the importance of these details.

Publication in any journal requires that claims about pre-CRISPR RNA processing are eliminated from the abstract or new structures and biochemistry are added to support these claims.

We thank the reviewer for the comments. As you mentioned, the processing mechanism via dimerization has not been sufficiently characterized based on the previous data. This is partly due to the low resolution of the original density map, which prevents the identification of the residues responsible for the pre-

crRNA processing. Therefore, we attempted to biochemically determine the active sites through *in vitro* processing assays. We introduced a 3'-Cy5-labeled 76-nucleotides pre-crRNA containing identical 20-nucleotides spacer sequences upstream and downstream of the 36-nucleotides repeat sequence (thus, pre-crRNA composed of spacer-repeat-spacer). We mixed the pre-crRNA with purified wild-type Cas12m2, as well as H269A, R270A, and H317A mutants, given that H269 and R270 in the WED domain were suggested to be involved in pre-crRNA processing in the previous study (Wu et al., 2022), and H317 is conserved as the RuvC active site. Consistent with our northern blotting analysis in the original paper, three variants exhibited *in vitro* pre-crRNA processing activity comparable to that of wild-type Cas12m2 (Fig. L1a). This result indicates that Cas12m2 processes its pre-crRNA in a unique manner, which is distinct from other Cas12 enzymes that use WED or RuvC domains for pre-crRNA processing. Our cryo-EM analysis revealed that the 5' end of the crRNA, which was previously reported to be processed, are surrounded by an additional density which aligns well with the REC1 domain, suggesting the presence of an active site responsible for pre-crRNA processing within the REC1 domain (Fig. L1b). We then substituted three conserved basic residues within the REC1 domain to alanine and performed *in vitro* processing assays. Among the mutants, H183A and H184A mutants have little or no processing activity *in vitro* (Fig. L1a), suggesting that H183 and H184 may be involved in pre-crRNA processing. However, since Cas12m2 exhibits very weak processing activity *in vitro* under various conditions (including buffer conditions, reaction times, and temperatures), further study is required to definitively identify H183 and H184 as catalytic residues. Therefore, as the reviewer suggested, we eliminated references to the dimeric

processing mechanism from the abstract and introduction sections, addressing it as a limitation of the study in the discussion section.

Figure L1. *In vitro* pre-crRNA processing analysis.

(a) *In vitro* pre-crRNA processing by the Cas12m2 and the variants. 3'-Cy5-labeled 76-nucleotides pre-crRNA

(composed of spacer-repeat-spacer) (1.6 μM) was mixed with Cas12m2 or Cas12m2 mutants (6.4 μM), and then incubated at 50°C for 30 min in a processing buffer (50 mM Tris-HCl pH 8.0, 100 mM NaCl, 10 mM MgCl_2). The reaction was stopped by the addition of quench buffer. The reaction products were then analyzed by 10% denaturing (7 M urea) PAGE.

(b) Cryo-EM density map (left) and model (center) of the Cas12m-crRNA binary complex. The additional three-helix-like density is indicated by a circle. The conserved basic three (R182A, H183A, and H184A) residues in the REC.1 domain are located close to the 5' end of the crRNA, which was previously reported to be processed (right).

MINOR SUGGESTIONS FOR INTENDED TO HELP IMPROVE CLARITY:

1. Ln 53: “In addition to the RuvC” seems incongruent with a sentence that describes differences not “additional” similarities. Consider define domain abbreviations used in Fig. 1. (e.g., TNB). Readers and directed to Fig. 1 well before they get to a definition for TNB in the text.

Thank you for the helpful comments. We changed the expression "in addition to" to "besides", as it was contextually inappropriate.

2. Ln 97: The authors state, “The REC lobe consists of a typical wedge (WED) but characteristic REC1 and REC2 domains.” Revise for clarity.

According to the comments, we have changed the text “The REC lobe consists of a typical wedge (WED) but characteristic REC1 and REC2 domains” to “The REC lobe consists of wedge (WED), REC1, and REC2 domains”, and explained the detail of each domain in the following sentences.

3. Fig. legend #1: The asterisk following numbering in NTS is not defined. The methods describing residues not included in the model are extremely difficult to follow. The gray dotted box, which is supposed to highlight these residues does, is not used in panel A for unmodeled residues and several of the unmodeled residues listed in the methods are not highlighted in the figure (e.g., 13–20 of the crRNA, 27 of the TS, etc). Consider using the word “through” in place of the “-“. This might make the sections a little more readable given the use of negative numbers (i.e., -9 through 5 rather than “(-9)-5”). Using the cartoon in panel “a” to highlight regions of the protein that are not included model due to the absence of sufficient density.

Thank you for the helpful comments. We added the explanation about the use of the asterisk symbol in the legend of Fig.1d and highlighted regions that are not included within the model in Fig.1c in the revised manuscripts. In addition, we replaced the symbol “-“ with the word “to” for describing a region of nucleic acids for the overall clarity. The unmodeled residues you pointed out (e.g., 13–20 of the crRNA, 27 of the TS) correspond to the model of the intermediate state of Cas12m2–crRNA–target DNA ternary complex, while Fig.1 (panel A as you may refer to it) represents the full R-loop state of the

ternary complex. In the full R-loop, nucleotides (−36) to (−30) of the crRNA, nucleotides (−9) to (−5) of the TS, and nucleotides 22 to 26 of the NTS were not included in the final model, and we highlighted these regions in revised figures (Fig. 1c and d).

4. Ln 117: “Unexpectedly, nucleotides dG(−3)–dC(−1) in the TS do not form base pairs with G18–C20 in the crRNA and instead re-hybridize with dG18–dC20* in the NTS, indicating that 17 nucleotides in the spacer sequence function as a guide segment in Cas12m2-mediated DNA recognition.” This is interesting and might be mechanistically important, but the authors do not explain the mechanistic basis for the “guide segment”. This makes the structure more observational rather than mechanistically insightful.*

Thank you for the comments. We changed the text “Unexpectedly, nucleotides dG(−3) to dC(−1) in the TS do not form base pairs with G18 to C20 in the crRNA and instead re-hybridize with dG18* to dC20* in the NTS, indicating that 17 nucleotides in the spacer sequence function as a guide segment in Cas12m2-mediated DNA recognition” to “Unexpectedly, nucleotides dG(−3) to dC(−1) in the TS do not form base pairs with G18 to C20 in the crRNA and instead re-hybridize with dG18* to dC20* in the NTS, suggesting that the 17-bp guide RNA–target DNA heteroduplex represents the optimal length for Cas12m2-mediated DNA recognition” in the revised manuscript.

5. Consider adding an atomic model to Movie S1. In the current version of the movie, it is hard to identify the where the truncated duplex in state II ends and how the extended duplex of state I is involved in the conformational change of REC2.

Thank you for the helpful comments. Accordingly, we have colored the maps derived from 3D variability analysis based on the domain configuration and have incorporated atomic models of both the intermediate state and the full R-loop state of the Cas12m2–crRNA–target DNA ternary complex into new Movie S1.

6. Consider enlarging labels in Fig. 4a, adding labels to major unannotated bands (e.g., at ~331). In Fig. 4b, do not see the pre-ordered A-form geometry highlighted in the text. Consider adding “sickle-like feature” to Ext. Data Fig. 2 that is currently labeled “kinked-coiled-coil” for consistency.

According to the comments, we enlarged the labels in Fig. 4a of the original manuscript and relocated the figure to Extended Data Fig. 8a in the revised manuscript. Furthermore, we performed *in vitro* transcription and translation assay, utilizing the third spacer as a probe for northern blotting to interpret the band near the marker representing ~331-nucleotide (in the original version, we used the first spacer as a probe). Although it is not easy to annotate all the bands, we annotated the band near the ~331-nucleotide marker in the revised figure (Extended Data Fig. 8a). Lastly, we incorporated the descriptive term "sickle-like architecture" in the relevant figures to enhance the clarity. As for the pre-ordered A-form geometry of crRNA, we eliminated the structural model of the Cas12m2–crRNA binary complex from figures for the overall clarity.

Reviewer #2:

In this manuscript, Omura et al. provided the cryo-EM structural elucidations of Cas12m2–crRNA binary and Cas12m2–crRNA–target DNA ternary complexes, claiming that the structural rearrangements of the REC lobe and the RuvC lid motif lead to a strong binding of the displaced NTS. Considering the null activity of Cas12m2 as to dsDNA breaks, the strong binding traits may confer adaptive immune defense strategy through robust transcriptional silencing. The authors also claim that an asymmetric homodimerization is preceded by the auto-cleavage of the cognate pre-crRNA, which is not fully supported by relevant evidence. In contrast to Cas12f, which compensates the PAM-distal recognition through dimerization, Cas12m2 is equipped with REC2 insertion for the purpose as is seen for Cas12a. The authors also provide a so-called “non-putative HDD motif” theory to explain why Cas12m2 is catalytically inactive as to dsDNA cleavage, which is partly associated with the loss of second Mg²⁺ ion. Finally, the authors provide some evolutionary insight into Cas12m2 with respect to CRISPR-Cas adaptive immunity through a structural comparison with TnpB. Overall, this manuscript provides unique structural features of Cas12m2 that were not reported hitherto for other Cas systems, but I cannot find any attractive points for the overall data provided in this manuscript and, in addition, have several major concerns as follows:

We thank the reviewer for the comments and would like to address concerns raised by the reviewer in the below point-by-point response.

1. The authors claimed that tetra-arginine residues (Arg111, Arg112, Arg126, Arg173) are involved in a non-base-specific recognition of NTS, thereby contributing to tight sequestration of NTS into the Arg-rich pocket. This may be a unique feature, but not a mechanism, of Cas12m for NT/NTS binding. The authors failed to provide elucidation as to how this unique feature affects the binding behavior of crRNA-Cas12m-dsDNA ternary complex and/or interference property of the RNP complex.

Thank you for the insightful comments. According to the reviewer’s comments, we conducted *in vitro* Surface Plasmon Resonance (SPR) analysis to address the DNA binding behavior of Cas12m2. We analyzed the interactions of wild-type Cas12m2, R111A, R112A, R126A, and R111A/R112A/R126A mutants with dsDNA fragments immobilized on a chip. Consistent with our structural observation of the

Cas12m2–crRNA–target DNA complex, target site binding of all mutants was significantly lower than that of wild-type Cas12m2 (Fig. L2a), indicating that Arg-rich pocket residues contribute to the strong DNA binding by Cas12m2. Additionally, we examined the interference property of Arg-pocket mutants of Cas12m2 in *E.coli*. We generated a target plasmid (pTarget-Operon) containing a bi-cistronic operon with two fluorescence reporter genes, *rfp* and *gfp*. *E.coli* cells harboring pTarget-operon and either pCRISPR-A1 (crRNA targeting the promoter region of the operon), pCRISPR-E1 (crRNA targeting the 3' UTR region of the operon) or pCRISPR-NT were made chemically competent and transformed with either pCas-Cas12m, pCas-dCas12m, or pCas-Cas12m-[x] (the four Ala-substitution mutants) (Fig. L2b, c). Although individual alanine substitutions of R111A, R112A, and R126A did not substantially influence the relative fluorescence of reporter GFP and RFP, R111A/R112A/R126A triple mutant shows considerably weaker transcriptional silencing activity compared to that of the wild-type Cas12m2 (Fig. L2c), indicating that Arg-rich pocket within the REC1 and REC2 domains contribute to the transcriptional silencing activity via the strong DNA binding activity of Cas12m2. We added these results

of *in vitro* SPR analysis (Fig. 2h) and *in vivo* transcriptional silencing assay (Fig. 3h) in the revised manuscript.

Figure L2. *In vitro* DNA target site binding and *in vivo* transcriptional silencing activity of Cas12m2.

(a) DNA binding of Cas12m2 mutants *in vitro*, measured by surface plasmon resonance spectroscopy, relative to wild type. Values shown are the mean \pm SEM of three independent experiments. Significant differences relative to wild type Cas12m2 are indicated by asterisks. (*): $p < 0.05$, (**): $p < 0.01$, (***) $p < 0.001$. Triple: R111A/R112A/R126A triple mutant.

(b) Schematic of bicistronic *rfp gfp* operon (pTarget-operon), including crRNA targeting sites in the promoter (A1) and at the end of the operon (3'-UTR; E1)

(c) Normalized RFP and GFP fluorescence of *E. coli* cultures expressing different Cas12m2 mutants, directed to either the promoter (pCRISPR-A1) or the 3'-UTR (pCRISPR-E1). Fluorescence is shown relative to average fluorescence of cultures expressing a non-targeting guide (pCRISPR-NT). Significant differences relative to Cas12m2 are indicated by asterisks. (*): $p < 0.05$, (**): $p < 0.01$, (***) : $p < 0.001$. dCas12m2: D485A mutant, Triple: R111A/R112A/R126A triple mutant.

2. The catalytically active D-E-D motif found in most Cas12a and TnpB nucleases was mutated into the H-D-D motif in Cas12m, which is suggested to explain why Cas12m shows null-activity toward dsDNA. If so, why don't you counter-mutate the motif into D-E-D and test if the mutated Cas12m restores the catalytic activity? This experiment could be the most crucial validation study on the phenomenon of catalytic loss of Cas12m2.

Thank you for the helpful comments. As suggested by the reviewer, we counter-mutated the non-canonical HDD motif into a canonical DED motif and examined the *in vitro* DNA cleavage activity. However, we failed to restore the catalytic activity of Cas12m2. We think that the activity of an enzyme is conferred through the presence of a catalytic site, which is intricately integrated into the elaborate molecular architecture of the entire protein, as opposed to being solely dependent upon the few residues constituting the active site. Therefore, in the case of Cas12m2, considering the structural differences extending beyond the residues of the active site, it is not unreasonable that merely counter-mutating the HDD catalytic center to the canonical DED motif would be insufficient to restore the DNA cleavage activity of Cas12m2.

3. According to the explanations by the authors, structural changes in the REC2 domain and the lid motif form the Arg-rich pocket and the opening of a RuvC magnesium ion binding site. This molecular trait is suggested to support a strong target DNA binding affinity of Cas12m2. This explanation is structurally quite plausible for addressing the strong binding affinity of Cas12m2. Nonetheless, it may be desirable to implement the mutation of those sites and to monitor the alterations in the binding affinity of the ternary complex.

Thank you for the helpful comments. As for the RuvC binding site, Wu *et al* demonstrated that the D485A mutant (HDD > HAD) has lower DNA binding affinity *in vitro*. In addition, we performed *in vivo* transcriptional silencing assay and the D485A mutant (dCas12m2) exhibited reduced transcriptional silencing activity (Fig. L2b, c). As for the Arg-rich pocket within the REC1 and REC2 domains, we conducted the *in vitro* SPR analysis and *in vivo* transcriptional silencing assay as described above, confirming that the Arg-rich pocket contributes to the transcriptional silencing activity via the strong DNA binding affinity of Cas12m2. We have added these results of *in vitro* SPR analysis (Fig. 2h) and *in vivo* transcriptional silencing analysis (Fig. 3h) in the revised manuscript.

4. When it comes to the auto-processing mechanism of crRNA, the authors failed to provide any supportive evidence, only providing a conjecture that dimerization of Cas12m2 might be the relevant mechanism. We cannot draw any conclusions from the current level of data, which significantly detract from the value of this study.

We thank the reviewer for the comments. As you mentioned, the processing mechanism via dimerization has not been sufficiently characterized based on the previous data. This is partly due to the low resolution of the original density map, which prevents the identification of the residues responsible for the pre-crRNA processing. Therefore, we attempted to biochemically determine the active sites through *in vitro* processing assays. We introduced a 3'-Cy5-labeled 76-nucleotides pre-crRNA containing identical 20-nucleotides spacer sequences upstream and downstream of the 36-nucleotides repeat sequence (thus, pre-crRNA composed of spacer-repeat-spacer). We mixed the pre-crRNA with purified wild-type Cas12m2, as well as H269A, R270A, and H317A mutants, given that H269 and R270 in the WED domain were suggested to be involved in pre-crRNA processing in the previous study (Wu *et al.*, 2022), and H317 is conserved as the RuvC active site. Consistent with our northern blotting analysis in the original paper, three variants exhibited *in vitro* pre-crRNA processing activity comparable to that of wild-type Cas12m2 (Fig. L1a). This result indicates that Cas12m2 processes its pre-crRNA in a unique manner, which is distinct from other Cas12 enzymes that use WED or RuvC domains for pre-crRNA processing. Our cryo-EM analysis revealed that the 5' end of the crRNA, which was previously reported to be processed, are surrounded by an additional density which aligns well with the REC1 domain, suggesting the presence of an active site responsible for pre-crRNA processing within the REC1 domain (Fig. L1b). We then

substituted three conserved basic residues within the REC1 domain to alanine and performed *in vitro* processing assays. Among the mutants, H183A and H184A mutants have little or no processing activity *in vitro* (Fig. L1a), suggesting that H183 and H184 may be involved in pre-crRNA processing. Therefore, we believe that these findings provide valuable insights into the unique processing mechanism by Cas12m. Nonetheless, as pointed out by the reviewer, further structural and biochemical studies are required to definitively identify H183 and H184 as catalytic residues. Thus, we eliminated references to the dimeric processing mechanism from the abstract and introduction sections, addressing it as a limitation of the study in the discussion section.

Figure L1. *In vitro* pre-crRNA

processing analysis.

(a) *In vitro* pre-crRNA processing by the Cas12m2 and the variants. 3'-Cy5-labeled 76-nucleotides pre-crRNA (composed of spacer-repeat-spacer) (1.6 μ M) was mixed with Cas12m2 or Cas12m2 mutants (6.4 μ M), and then incubated at 50°C for 30 min in a processing buffer (50 mM Tris-HCl pH 8.0, 100 mM NaCl, 10 mM MgCl₂). The reaction was stopped by the addition of quench buffer. The reaction products were then analyzed by 10% denaturing (7 M urea) PAGE.

(b) Cryo-EM density map (left) and model (center) of the Cas12m–crRNA binary complex. The additional three-helix-like density is indicated by a circle. The conserved basic three (R182A, H183A, and H184A) residues in the REC.1 domain are located close to the 5' end of the crRNA, which was previously reported to be processed (right).

5. The last section of this manuscript is handling the evolutionary aspect of Cas12m through the structural comparison of TnpB. However, the only additive is that the acquisition of REC2 and La2 helix elongation is associated with the recognition of longer target sequence and an increased PAM distal binding. I think this level of interpretation does not expand the scope of our understanding toward TnpB-Cas12a evolutionary axis.

Thank you for the comments. As you pointed out, our statement in this paper, suggesting that Cas12m2 evolved from TnpB through domain insertion and expansion, may not comprehensively expand the scope of our understanding of the TnpB-Cas12a evolutionary axis. However, the key point we wish to emphasize is that the only insertion of the REC2 domain into the TnpB core enables TnpB to engage in the CRISPR-Cas immunity through the recognition of the longer guide RNA–target DNA heteroduplex. This insight may not be attainable when comparing other Cas12 enzymes, such as Cas12a or Cas12b, which exhibit low structural similarity with TnpB besides the RuvC domain. The observation that Cas12m2, while maintaining a remarkable resemblance to TnpB in its basic structure, incorporates the inserted REC2 domain and subsequently recognizes elongated target DNA sequences, could constitute pivotal evidence suggesting that the minimal domain insertion was sufficient for TnpB to become engaged in an adaptive immunity during its initial evolutionary phase.

6. One of the core points of this manuscript would be that the tight DNA binding mechanism may confer robust transcription silencing option, thereby empowering an adaptive immune defense tool to the microorganism harboring CRISPR-Cas12m system. This should be supported by any minimal level of biochemical or in vivo validation study, but I can't find any relevant experiment. One of the tactics would be that you incorporate a viral protospacer sequence into the CRISPR loci of Mycolicibacterium

mucogenicum and test if the viral target-harboring bacteria could gain survival advantage against viral attack. Or an alternative approach could be welcomed.

We appreciate your perspective comments. To validate the transcriptional silencing activity mediated by the strong DNA binding of Cas12m2 through the Arg-rich pocket and the RuvC DNA binding site, we performed an *in vivo* transcriptional silencing assay (described above). The wild-type Cas12m2, targeting both the promoter and the 3' UTR region of the reporter gene, showed robust transcriptional silencing activity *in vivo* (Fig. L2b, c). In contrast, the D485A mutant (dCas12m2; RuvC DNA binding site is disrupted) exhibited diminished activity when targeting the 3' UTR region of the reporter gene, and the R111A/R112A/R126A triple mutant (Arg-rich pocket within the REC1 and REC2 domains is disrupted) targeting both the promoter and the 3' UTR region displayed considerably weaker transcriptional silencing activity (Fig. L2b, c). Although we cannot directly examine whether Cas12m2 offers an advantage against viral attack for *Mycolicibacterium mucogenicum* due to our technical limitation, we believe that the *in vivo* assay described here sufficiently address the transcriptional silencing activity of Cas12m2 through tight target DNA binding.

Decision Letter, first revision:

11th May 2023

Dear Prof. Nureki,

Thank you for submitting your revised manuscript "Mechanistic and evolutionary insights into a type V-M CRISPR-Cas effector enzyme" (NSMB-A47332A). It has now been seen by the original referees and their comments are below. The reviewers find that the paper has improved in revision, and therefore we'll be happy in principle to publish it in Nature Structural & Molecular Biology, pending minor revisions (textual clarifications, toning down, and providing some controls as supporting data) to satisfy the referees' final requests and to comply with our editorial and formatting guidelines.

Sincerely,

Dimitris Typas
Associate Editor
Nature Structural & Molecular Biology
ORCID: 0000-0002-8737-1319

Reviewer #1 (Remarks to the Author):

The revised manuscript addresses the primary concerns. I recommend the paper for publication after the authors address a few additional concerns/suggestions.

- Why are the binding data plotted as “relative binding” rather than KD? SPR should provide on and off rates, and a ratio of the two is the KD. Reporting the KD and/or adding a bit more sophisticated explanation of the SPR data seems particularly relevant for a DNA binding protein whose function is directly related to DNA binding affinity. It is not clear what “[−]” represents on the y-axis. Even if the authors chose to stick with “relative binding” in the main text, the extended data should provide more information/detail about the data generated using SPR. I understand that it might not be reasonable to estimate KD for the mutants, but this should be done for the WT, at least.
- Add SEC and SDS gels of the purified proteins (i.e., WT and each of the mutants).

Minor suggestions:

- Ln 115 recommend revising for clarity. The crRNA scaffold comprises a pseudoknot (PK), stem, and loop regions. The 5′ end of the crRNA (G(−36) to U(−30)) is disordered (Fig. 1d, e and Extended Data Fig. 2f).
- Double check table 1 to make sure more protein atoms are modeled in the low-resolution structure than the high-resolution structure.
- Consider adding a few more pertinent details to table 1 (data collection software, image processing package, 3DFSC sphericity, etc).
- Ln 531: change “Ribonucleotide protein” to “Ribonucleoprotein”
- In some places the authors use a dash (ln 460: 3.73-Å) and other places they don’t (ln 458: 1.66 Å). Please check for other inconsistencies throughout.

Reviewer #2 (Remarks to the Author):

In this manuscript, the authors gave their best shot to address the questions raised previously by conducting several biochemical works. This manuscript was significantly improved in terms of solidness and integrity of data. I’d like to note one remaining question as follows:

The authors attempted to address the questions raised previously as follows. However, the answer the authors provided is likely to prove that the replacement of D-E-D motif with H-D-D one may not be a major cause for the null activity of Cas12m. As the authors stated, there could be more reasons why Cas12m lost the catalytic activity besides the H-D-D motif. This issue should need to be fully addressed.

Author Rebuttal, first revision:

Responses to reviewers’ comments

Reviewer #1:

The revised manuscript addresses the primary concerns. I recommend the paper for publication after the authors address a few additional concerns/suggestions.

We thank the reviewer for the positive comments. We would like to address the concerns raised by the reviewer in the following point-by-point responses.

1. Why are the binding data plotted as “relative binding” rather than KD? SPR should provide on and off rates, and a ratio of the two is the KD. Reporting the KD and/or adding a bit more sophisticated explanation of the SPR data seems particularly relevant for a DNA binding protein whose function is directly related to DNA binding affinity. It is not clear what “[−]” represents on the y-axis. Even if the authors chose to stick with “relative binding” in the main text, the extended data should provide more information/detail about the data generated using SPR. I understand that it might not be reasonable to estimate KD for the mutants, but this should be done for the WT, at least.

Thank you for the helpful comment. The binding of Cas12m2 to a 50 bp dsDNA containing a target site displays multiphasic kinetics (see also Wu et al., Mol Cell 2022), which may be due to (a combination of) low affinity non-target site binding, a conformational change and/or the quality of the CRISPR-RNA. Thus, the complete set of sensorgrams cannot be properly fitted with a simple 1:1 binding model or a 2-state model and an accurate KD could not be determined. Nonetheless, an estimation of the KD for the wild-type Cas12m2 was obtained in the low nM range upon fitting a subset of sensorgrams (8-63 nM) to a two-state model, and we have included this analysis in Extended Data Figure 3 in the revised manuscript. For the remaining mutants, the binding capacity was so low (even at the highest concentration of 500 nM RNP complex, see extended figure 3b) that this kinetic analysis could not be performed. Therefore, instead of reporting a KD for the mutants, we reported the relative binding from the results of the SPR analysis. According to the comment, we have removed the [−] on the y-axis of Fig. 2h in the revised manuscript.

2. Add SEC and SDS gels of the purified proteins (i.e., WT and each of the mutants).

Thank you for the comment. We have added SEC profiles and SDS gels of the purified proteins as Extended Data Fig. 4 in the revised manuscript.

MINOR SUGGESTIONS:

3. Ln 115 recommend revising for clarity. The crRNA scaffold comprises a pseudoknot (PK), stem, and loop regions. The 5' end of the crRNA (G(-36) to U(-30)) is disordered (Fig. 1d, e and Extended Data Fig. 2f).

According to the comment, we have changed the text “The crRNA scaffold comprises pseudoknot (PK), stem, and loop regions, but the 5' end of the crRNA (G(-36) to U(-30)) is disordered” to “The crRNA scaffold comprises a pseudoknot (PK), stem, and loop regions. The 5' end of the crRNA (G(-36) to U(-30)) is disordered.” for clarity.

4. Double check table 1 to make sure more protein atoms are modeled in the low-resolution structure than the high-resolution structure.

Thank you for the comment. The structure of the Cas12m-crRNA binary complex deposited in the PDB for this study is registered as a dimeric structure. Therefore, more protein atoms are modeled in the Cas12m-crRNA binary complex than in the Cas12m-crRNA-target DNA ternary complex.

5. Consider adding a few more pertinent details to table 1 (data collection software, image processing package, 3DFSC sphericity, etc).

Thank you for the helpful comment. We have added some statistics to Table 1 in the revised manuscript, including data collection software, image processing package, and 3DFSC analysis.

6. Ln 531: change “Ribonucleotide protein” to “Ribonucleoprotein”.

We have changed the term “Ribonucleotide protein” to “Ribonucleoprotein” in the revised manuscript.

7. In some places the authors use a dash (ln 460: 3.73-Å) and other places they don't (ln 458: 1.66 Å). Please check for other inconsistencies throughout.

Thank you for the helpful comment. We have removed the dashes in the revised manuscript for consistency.

Reviewer #2:

Remarks to the Author: In this manuscript, the authors gave their best shot to address the questions raised previously by conducting several biochemical works. This manuscript was significantly improved in terms of solidness and integrity of data. I'd like to note one remaining question as follows:

The authors attempted to address the questions raised previously as follows. However, the answer the authors provided is likely to prove that the replacement of D-E-D motif with H-D-D one may not be a major cause for the null activity of Cas12m. As the authors stated, there could be more reasons why Cas12m lost the catalytic activity besides the H-D-D motif. This issue should need to be fully addressed.

We thank the reviewer for the positive comments. The cryo-EM structure of the Cas12m2–crRNA–target DNA ternary complex revealed that the non-canonical HDD-type RuvC catalytic triad coordinates only one magnesium ion, thus abolishing the DNA cleavage activity of Cas12m2. However, there could be more reasons why Cas12m2 lost the DNA cleavage activity besides the non-canonical HDD-type RuvC triad. To fully understand the structural mechanisms behind the loss of cleavage activity in Cas12m2, it would be critical to compare the structures of Cas12m2 with those of clade 3 and clade 4 Cas12m enzymes, which possess the canonical DED-type RuvC catalytic center and might cleave target DNA. According to the reviewer's suggestion, we have included brief statements about this issue in the discussion section of the revised manuscript.

Final Decision Letter:

Message 22nd Jun 2023

:

Dear Professor Nureki,

We are now happy to accept your revised paper "Mechanistic and evolutionary insights into a type V-M CRISPR-Cas effector enzyme" for publication as a Article in Nature Structural & Molecular Biology.

Your paper will be published online soon after we receive proof corrections and will appear in print in the next available issue. You can find out your date of online publication by contacting the production team shortly after sending your proof corrections. Content is published online weekly on Mondays and Thursdays, and the embargo is set at 16:00 London time (GMT)/11:00 am US Eastern time (EST) on the day of publication. Now is the time to inform your Public Relations or Press Office about your paper, as they might be interested in promoting its publication. This will allow them time to prepare an accurate and satisfactory press release. Include your manuscript tracking number (NSMB-A47332B) and our journal name, which they will need when they contact our press office.

About one week before your paper is published online, we shall be distributing a press release to news organizations worldwide, which may very well include details of your work. We are happy for your institution or funding agency to prepare its own press release, but it must mention the embargo date and Nature Structural & Molecular Biology. If you or your Press Office have any enquiries in the meantime, please contact press@nature.com.

If you have not already done so, we strongly recommend that you upload the step-by-step protocols used in this manuscript to the Protocol Exchange. Protocol Exchange is an open

online resource that allows researchers to share their detailed experimental know-how. All uploaded protocols are made freely available, assigned DOIs for ease of citation and fully searchable through nature.com. Protocols can be linked to any publications in which they are used and will be linked to from your article. You can also establish a dedicated page to collect all your lab Protocols. By uploading your Protocols to Protocol Exchange, you are enabling researchers to more readily reproduce or adapt the methodology you use, as well as increasing the visibility of your protocols and papers. Upload your Protocols at www.nature.com/protocolexchange/. Further information can be found at www.nature.com/protocolexchange/about.

Please note that *Nature Structural & Molecular Biology* is a Transformative Journal (TJ). Authors may publish their research with us through the traditional subscription access route or make their paper immediately open access through payment of an article-processing charge (APC). Authors will not be required to make a final decision about access to their article until it has been accepted. [Find out more about Transformative Journals](https://www.springernature.com/gp/open-research/transformative-journals)

Authors may need to take specific actions to achieve [compliance with funder and institutional open access mandates](https://www.springernature.com/gp/open-research/funding/policy-compliance-faqs). If your research is supported by a funder that requires immediate open access (e.g. according to [Plan S principles](https://www.springernature.com/gp/open-research/plan-s-compliance)) then you should select the gold OA route, and we will direct you to the compliant route where possible. For authors selecting the subscription publication route, the journal's standard licensing terms will need to be accepted, including [self-archiving policies](https://www.springernature.com/gp/open-research/policies/journal-policies). Those licensing terms will supersede any other terms that the author or any third party may assert apply to any version of the manuscript.

Sincerely,

Dimitris Typas
Associate Editor
Nature Structural & Molecular Biology
ORCID: 0000-0002-8737-1319
